# Physical Training Protocols for Improving Dyspnea and Fatigue in Long COVID: A Systematic Review with Meta-Analysis

**DOI:** 10.3390/healthcare13151897

**Published:** 2025-08-04

**Authors:** Lisa Fernanda Mazzonetto, Jéssica Fernanda Correa Cordeiro, Igor Massari Correia, Alcivandro de Sousa Oliveira, Chimenny Moraes, Joana Brilhadori, Eurípedes Barsanulfo Gonçalves Gomide, Michal Kudlacek, Dalmo Roberto Lopes Machado, Jeferson Roberto Collevatti dos Anjos, André Pereira dos Santos

**Affiliations:** 1School of Physical Education and Sport of Ribeirão Preto, University of São Paulo, Ribeirão Preto 13566-590, Brazil; lisafernanda20@hotmail.com (L.F.M.); alcivandroox@hotmail.com (A.d.S.O.); jo.joana.br2@gmail.com (J.B.); dalmo@usp.br (D.R.L.M.); collevatti.jeferson@usp.br (J.R.C.d.A.); andrepereira.educa@gmail.com (A.P.d.S.); 2Study and Research Group in Anthropometry, Training and Sport, School of Physical Education and Sport of Ribeirão Preto, University of São Paulo, Ribeirão Preto 13566-590, Brazil; igormassari@usp.br (I.M.C.); chimenny.mor@gmail.com (C.M.); euripedesgomide@claretiano.edu.br (E.B.G.G.); 3Research Center in Physical Activity, Health and Leisure (CIAFEL), Faculty of Sports, University of Porto, 4200-319 Porto, Portugal; 4College of Nursing of Ribeirão Preto, University of São Paulo, Ribeirão Preto 13566-590, Brazil; 5Claretiano—University Center, São Paulo 14300-970, Brazil; 6Faculty of Physical Culture, Palacký University Olomouc, 771 46 Olomouc, Czech Republic; michal.kudlacek@upol.cz; 7School of Education and Communication (ESEC), University of Algarve, Penha Campus, 8005-139 Faro, Portugal; 8Department of Child, Family and Population Health Nursing, University of Washington, Seattle, WA 98195, USA

**Keywords:** post-COVID-19, exercise, persistent symptoms, shortness of breath

## Abstract

**Objective:** This study aimed to evaluate physical training protocols for alleviating long COVID symptoms, especially dyspnea and fatigue, through a systematic review with meta-analysis. **Method:** Data were collected from EMBASE, LILACS, PubMed, Scopus, CINAHL, Web of Science, and grey literature (Google Scholar, medRxiv). Studies evaluating dyspnea and/or fatigue before and after physical rehabilitation, using validated questionnaires, were included. Studies lacking pre- and post-assessments or physical training were excluded. Two reviewers independently extracted data on intervention type, duration, frequency, intensity, and assessment methods for dyspnea and fatigue. Bias risk was evaluated using the Cochrane tool. **Results:** Combined methods, such as respiratory muscle training with strength and aerobic exercise, were common for long COVID symptoms. Aerobic exercise notably improved dyspnea and/or fatigue. Among 25 studies, four had a low risk of bias. Meta-analysis of two studies found no significant reduction in fatigue. **Conclusion:** Combined training methods, particularly aerobic exercise, alleviate dyspnea and fatigue in long COVID. More high-quality studies are needed to confirm these findings.

## 1. Introduction

In May 2023, the World Health Organization declared the end of the Public Health Emergency of International Concern (PHEIC) regarding COVID-19. However, clinical outcomes of post-COVID remain prevalent. It is estimated that globally, 25 to 30% (~144 million) of individuals diagnosed with symptomatic COVID-19 will experience long-term sequelae after infection [1,2]. The prevalence of this condition is still unknown, and reports suggest that associated comorbidities, such as obesity and pre-existing respiratory problems, may be potential factors for some of the sequelae [1,3]. The most frequent persistent symptoms include fatigue, dyspnea, headaches, chest and musculoskeletal pain, loss of smell and/or taste, exercise intolerance, and memory loss [4,5]. Persistent symptoms of COVID-19, referred to as long COVID, involve multiple organ systems, including the respiratory, neurological, cardiovascular, gastrointestinal, endocrine, and musculoskeletal systems as well as biopsychosocial mechanisms such as neuroinflammation and mitochondrial dysfunction [5]. Due to the diversity of symptoms and the extent of affected organ systems, diagnosing the disease has been challenging. A characteristic feature of these persistent symptoms is their duration of twelve weeks or more following the acute phase of the disease [6]. Studies have also used continuous symptoms lasting twelve weeks or longer after infection as a criterion for inclusion in the sample [3,5,7]. Respiratory symptoms are a significant set of defined symptoms, identified through a collection of data from studies on persistent long COVID symptoms, including dyspnea, respiratory pattern disturbances, persistent cough, pulmonary fibrosis, and thromboembolic disease. Among these symptoms, fatigue is also commonly reported, resembling myalgic encephalomyelitis. These symptoms require specific evaluation and interventions targeted toward these conditions [3,8].

Respiratory exercises and chest physiotherapy promote the health of patients who are discharged, but little is known about how to approach long COVID [6]. People with different types of continuous symptoms struggle with daily activities [5]. Dyspnea can cause extreme fatigue, making simple tasks such as walking, climbing stairs, or even speaking challenging. This affects the autonomy and overall well-being of individuals with long COVID. Similarly, chronic fatigue, characterized by deep exhaustion that does not improve with rest, hinders daily tasks and even social activities, negatively impacting both physical and mental health. The pathophysiology of long COVID is still not fully understood, highlighting the need for intervention studies with various physical training protocols that provide the best results for those affected by the sequelae of COVID-19. The specificity of exercises may help alleviate the persistent symptoms of the disease. To date, there is no mapping of physical training protocols and their effectiveness in alleviating persistent symptoms of long COVID, particularly concerning dyspnea and fatigue. Understanding the different physical training protocols applied to people in this condition may point to a better approach for prescribing physical exercise to mitigate dyspnea and fatigue during long COVID, thereby improving the quality of life for these patients. Therefore, the aim of this systematic review and meta-analysis was to map the training protocols used to alleviate symptoms of dyspnea and fatigue and assess their effects in individuals with long COVID.

## 2. Materials and Methods

This systematic review and meta-analysis was developed and described following the guidelines proposed by the Preferred Reporting Items for Systematic Reviews and Meta-Analyses (PRISMA). The review protocol was registered on the Open Science Framework (OSF) platform under the DOI registration 10.17605/OSF.IO/MX86G on 31 October 2023.

The research question (“What types of physical training are adopted by people with long COVID to alleviate symptoms of dyspnea and fatigue resulting from the disease?”) was formulated using the PICO strategy, as follows: [1] (P) Population: individuals living with long COVID; [2] (I) Intervention: physical training protocols; [3] (C) Comparison: not applicable; [4] (O) Outcome: improvement of dyspnea and fatigue symptoms caused by COVID-19.

In this review, primary experimental studies (clinical trials, quasi-experimental studies, case–control studies, case series, and case reports) that applied a physical training protocol for individuals with dyspnea and fatigue symptoms due to long COVID were included. Secondary studies (other reviews), editorials, books, book chapters, guidelines, expert opinion articles, dissertations, theses, and conference abstracts on the topic were excluded. The search for articles was conducted in October 2023 using the following databases: Excerpta Medica Database (EMBASE), Latin American Literature in Health Sciences (LILACS), National Library of Medicine via the PubMed portal, Scopus, Cumulative Index to Nursing & Allied Health Literature (CINAHL) via EBSCO, and Web of Science. Additionally, grey literature was considered, including Google Scholar and medRxiv.

The search strategy used was formulated through the combination of controlled terms and keywords related to the topic (mainly: post-acute COVID-19 syndrome; exercise; fatigue; dyspnea), respecting the specificities of each database (Table 1).

The identified studies were initially exported to EndNote Basic^®^ 21 software to eliminate duplicates across the databases. Then, the studies (without duplicates) were exported to Rayyan software, where analysis and selection were conducted based on eligibility criteria in two phases. In Phase 1, analysis and selection were performed by reading the titles and abstracts of all studies. In Phase 2, analysis and selection were performed by reading the full texts of only the studies selected in the previous phase.

A total of 1328 articles were analyzed based on their titles and abstracts. Of these, 1239 studies did not meet the inclusion criteria and were excluded. When there were doubts about a study’s eligibility based solely on the title and abstract, the full text was obtained and assessed for suitability. The search strategy was developed by two researchers (LM and JC), who consulted a third researcher (JB) in case of disagreement. After the initial analysis of the titles and abstracts, 89 articles were identified as potentially eligible, and a thorough reading of their full texts was conducted. As a result of this process, 25 articles were included in the review, as illustrated in the flowchart (Figure 1).

The data from the selected studies were extracted and compiled into a single characterization table, containing the following information: Study Characteristics: design, objective, country where the study was conducted, intervention duration, assessment tools used to evaluate dyspnea and/or fatigue pre- and post-intervention, and training protocols; Participant Characteristics: sample size, gender, average age; Main Outcomes: clinical outcomes related to dyspnea and/or fatigue symptoms following interventions with different physical training protocols aimed at mitigating dyspnea and fatigue symptoms resulting from long COVID.

The risk of bias assessment of the included studies was conducted using the Cochrane Collaboration Risk of Bias Tool (RoB 2) for randomized studies [9]. To synthesize the risk of bias analysis, the Cochrane Collaboration’s Review Manager 5 (RevMan 5.4) tool was used, allowing the construction of the risk of bias summary and graph [10]. It is important to note that all processes of study analysis and selection, data extraction, and risk of bias evaluation of eligible studies were performed independently and blindly by two researchers. Any conflicts arising during the process were resolved by a third researcher.

The meta-analysis was conducted using Review Manager software (RevMan Version 5.4.1). Due to considerable clinical heterogeneity among the included studies, the analysis had some limitations. We used the I^2^ statistic to assess the degree of heterogeneity, with percentages indicating the magnitude: 25% for low, 50% for moderate, and 75% for high heterogeneity. According to this scale, when the I^2^ value was equal to 50%, a random-effects model was adopted. The mean difference with a 95% confidence interval was employed. To illustrate the effects of the interventions on fatigue, a forest plot was generated.

## 3. Results

After completing the selection process, 25 articles were identified that answered the guiding question of this study. A summary of the main characteristics of each study is presented in Table 2, and additional details are provided in the Appendix A to offer a more comprehensive description of each included study.

### 3.1. Physical Training Protocols Used

The most commonly used physical training protocols were combinations of two or more methods. Among these, the most frequently used protocol combined respiratory muscle training (RMT) with aerobic exercise and strength training [11,12,25,33,34]. The second most common combination was strength training and aerobic exercise [15,21,22,26] the third was a combination of strength training, stretching, aerobic exercise, and RMT [24,27]. Additionally, some studies used RMT combined with functional training, stretching, aerobic training, strength training, and multicomponent training, with the latter including yoga-based respiratory exercises. Other studies applied training in isolation, without combining two or more methods, including Tai Chi, aerobic training, aquatic training, multicomponent training, strength training, RMT, and concurrent training (strength and aerobic training in the same session).

### 3.2. Effect of Training Protocols on Fatigue and Dyspnea

Of the 17 studies that assessed dyspnea, 11 showed significant improvement after physical exercise rehabilitation [17,18,23,24,25,27,28,29,30,32,35]. Of the 20 studies that assessed fatigue, 10 showed significant improvement after physical training [13,14,15,24,26,27,28,30,32,33].

A study [34] showed periods of worsening for both fatigue and dyspnea as a fluctuating pattern after rehabilitation with physical training. It presented clinical improvement in the variables of muscle strength, muscle power, and physical function. However, in some sessions, an increase in the intensity of fatigue and dyspnea symptoms was reported after excessive effort, with complaints of these symptoms persisting until the end of rehabilitation. The physical training protocol used was a combination of aerobic training, strength training, and inspiratory muscle training (a comprehensive use of aerobic training methods and strength training) [34]. The study by [28] presented a sample divided into four groups, each undergoing a different intervention. The interventions were: [1] concurrent training, ref. [2] inspiratory muscle training (IMT), ref. [3] a combination of concurrent training with IMT, and [4] a self-directed program following WHO recommendations, which was a more general and comprehensive program. Among these, the control group (WHO recommendations) showed no changes in fatigue or dyspnea, unlike the other groups. Significant changes in fatigue and dyspnea were observed with the concurrent training protocols and the combination of concurrent training with IMT [28].

Thus, a variation in the results regarding the effect on fatigue and/or dyspnea can be observed in the included studies, which may be due to the different physical training methods applied.

### 3.3. Assessment of Dyspnea

Three studies used the Modified Borg Scale for the assessment of dyspnea [22,25,35]; seven studies used the Medical Research Council (mMRC) Scale of 0–4 points [19,23,28,29,31,32,34]; One study used the Acute Respiratory Distress Syndrome (ARDS) Berlin criteria scale (based on the degree of hypoxia—PaO_2_/FiO_2_) [12] one study used the London Chest Activity of Daily Living Scale—Spanish version (LCADL) [14]; one study used the 12-Point Dyspnea Scale for COPD and pulmonary hypertension (D-12) [18]; one study used the Mahler Dyspnea Scale (7–10 points) [17]; one study used the Post-COVID Functional Status Scale (PCFS) [30]. Seven studies did not assess dyspnea, only fatigue.

### 3.4. Assessment of Fatigue

Four studies used the Fatigue Severity Scale (FSS) [13,20,29,30,32]; three studies used the Fatigue Assessment Scale (FAS) [14,19,31]; one study used the Fatigue Impact Scale (FIS) [22]; two studies used the Chalder Fatigue Scale (CFS-11) [26,28]; one study used the Fatigue Resistance Index (FRI) [11]; one study used the Brief Fatigue Inventory (BFI) along with the Post-COVID Functional Status Scale (PCFS) [15]; one study used the Fatigue Symptoms Questionnaire (FSQ) [16]; two studies used the Modified Borg Scale (PSEm) [33,35]; one study used the Fatigue Pictogram [21]. Five studies did not assess fatigue, only dyspnea.

Three studies used the same assessment instrument for both dyspnea and fatigue: the Modified Borg Scale, [35] the SF-36 questionnaire for physical function limitation and energy and fatigue [27] and the PFSDQ-M (Modified Dyspnea and Pulmonary Function Status Questionnaire) [24].

### 3.5. Methodological Quality of Included Articles

In Figure 2, most studies show an uncertain or high risk of bias. Only four studies were identified with a low risk of bias [14,16,28,32].

### 3.6. Meta-Analysis

After analyzing the articles included in the study, a meta-analysis was possible with two studies that investigated the effect of physical training on fatigue. This is because only these articles provided pre- and post-intervention data, along with a control group, and reported means and standard deviations to assess fatigue. For other outcomes and articles included, it was not feasible to perform a meta-analysis due to heterogeneity in measurement methods, different questionnaires used, and incomplete information.

The meta-analysis of the two intervention studies, one with low risk of bias 25 and the other with high risk of bias 30, revealed that the physical training protocol had no significant effect on reducing fatigue among groups of people with long COVID who received the intervention compared to the control group (total participants in the intervention: 37, in the control group: 38; 95% CI: 15.42 to 8.77; I^2^: 89%) (Figure 3).

## 4. Discussion

This study explored which physical training protocols are used for people with long COVID to alleviate symptoms of dyspnea and fatigue, the most commonly reported symptoms of long COVID [36]. Various physical training protocols were identified. Predominantly, combined training of two or more modalities, such as respiratory muscle training (RMT) with strength training and aerobic training, was the most utilized. This was followed by combinations of strength training and aerobic training, and the combination of strength training, stretching, aerobic training, and RMT. These findings corroborate with review studies [1] that discuss possible mechanisms, risks, and recovery from long COVID based on a physiology society conference with discussions on the topic, and [2] that discuss characteristics and treatment strategies for long COVID with a focus on exercise intolerance. They highlight the diverse strategies being employed and the importance of these different approaches in prescribing focused and adapted exercise for this population to reduce the symptom burden of the disease [1,2].

Of the articles included in this review, those that showed significant improvement after the physical training protocol were mostly protocols that combined two or more training methods. The studies that assessed and showed significant improvement in fatigue after training were [14,15,26,33]. Those that showed significant improvement in dyspnea were [17,18,23,25,29,35]. The studies [24,27,28,30,32] assessed both fatigue and dyspnea, with significant improvement in both symptoms, all of which involved combined training methods.

Although these studies have the same objective of applying physical training to alleviate the persistent symptoms of long COVID, the heterogeneity among the studies, such as sample size, patient characteristics (age range or even associated comorbidities), the assessment tool used, and the intervention itself—such as the duration of the intervention, the scope of the methods applied—creates variability. Some of these protocols are more general, such as self-directed programs with freer exercises, while others are more specific with supervision, or use tele-rehabilitation with structured exercise application, presenting a more specialized intervention, as seen in the studies of [17,18,21,28]. These methodological differences may influence the effects of physical training, resulting in either significant improvement or no improvement at all.

Aerobic exercise was present in most of the combined methods [14,15,18,23,24,26,27,28,29,30,33,35]. This indicates that aerobic exercise plays a significant role in the rehabilitation of the disease. Additionally, one study evaluated fatigue using aerobic training alone and reported significant improvement following the intervention [13]. This finding is consistent with the broader scientific literature supporting the benefits of aerobic exercise in addressing physical deconditioning and enhancing cardiorespiratory function. Aerobic training may contribute to the management of long COVID symptoms, particularly fatigue and dyspnea, through several physiological mechanisms, including improved mitochondrial function, reduced systemic inflammation, modulation of the autonomic nervous system, and increased cardiorespiratory and muscular efficiency [2,37].

A systematic review [2] presented studies evaluating the impact of physical training on patients with long COVID and advocates for exercise prescription focused on cardiac adaptations for individuals in this condition. A systematic review with meta-analysis showed significant improvement in the 6-min walk test (6MWT) (peak VO2) and the distance covered in the 6MWT (distance traveled) in young and middle-aged adults (n = 682) who underwent rehabilitation with aerobic exercise combined with strength and respiratory training, concluding that the applied training protocol is effective in improving cardiorespiratory fitness in individuals with long COVID [38].

With the multisystemic nature of long COVID and the variety of persistent symptoms, studies have highlighted the importance of a comprehensive multidisciplinary treatment approach that encompasses both physical and cognitive aspects [5,7]. Despite this multidisciplinary approach, it is crucial that each part of the rehabilitation process be specialized, focusing on the needs of individuals suffering from the sequelae [2]. The studies in this review assessed different physical training protocols for their effectiveness in mitigating the sequelae of long COVID. Among the variables and symptoms addressed, such as cardiopulmonary function, muscle strength, exercise tolerance, fatigue, dyspnea, mental confusion, and others, the interventions were also evaluated for safety, feasibility, tele-rehabilitation programs, self-managed training, and comparisons of different training protocols, all with the aim of analyzing the applied interventions to improve long COVID conditions.

A meta-analysis was conducted to evaluate the effect of the training protocol on reducing fatigue [13,28] which revealed that the physical training adopted had no significant effect on this symptom. Not all studies included in this review provided subdivision of groups by sex or age range (only reported as ≥18 years), but for those that did address the symptom of fatigue, the average age was 44.6 years. One meta-analysis showed that age affected the cardiopulmonary response of people with long COVID after rehabilitation with physical training, with responses being more favorable in younger adults compared to older adults or the elderly [38]. Age could be a possible explanation for the non-significant effect of training on fatigue in our analysis.

Additionally, the duration of the interventions in these studies (one of 12 weeks and the other of 8 weeks) may not have been sufficient to effect a change in fatigue. There is no commonly accepted definition for studies measuring between 3 and 24 weeks and the duration of symptoms [1]. People with long COVID exhibit significantly elevated inflammatory markers for several months, and there are multiple physiological pathways through which the immune system can influence the nervous system, potentially involving neuromuscular mechanisms that contribute to fatigue [39,40].

The sample size (n) of two studies [13,28] included in the meta-analysis may have influenced the non-significant effect of physical training on fatigue alleviation, which shows an improvement with training, but without statistical significance. The heterogeneity of the studies prevented a larger sample size for this analysis. The lack of standardization in the questionnaires used to measure fatigue and/or dyspnea makes it difficult to interpret the results, potentially leading to incompatible variables, which weakens the analysis and negatively impacts the comparability and reliability of the data analysis, undermining the integrity and quality of the conclusions. Therefore, only two studies were included in the meta-analysis.

Some studies discuss uncontrolled studies or methodological flaws (such as whether they were randomized or not, blinding of participants and researchers, missing data and control groups) and even the difficulty of understanding the pathophysiology of long COVID and changes in behavior due to the pandemic. These factors hinder the conduct of a meta-analysis with the necessary rigor [1,2,41]. At the same time, they emphasize the importance of continuous research to understand risk factors, diagnostic mechanisms, and treatment of long COVID. Understanding the disease’s pathophysiology can also support other post-viral syndromes [1,41]. In line with our results, despite the uncertain or high risk of bias in most of the studies, their development is essential. The studies included in this review, whether experimental or quasi-experimental, are crucial for guiding the rehabilitation of long COVID based on evidence. Current guidelines for individuals with long COVID are primarily based on observational data and expert opinion [1,36,41].

The results of this mapping reveal that interventions are based on combined method protocols aiming to maximize the effects of physical training in alleviating the persistent symptoms of the disease. In some studies, such as those by [42] it was demonstrated that inspiratory muscle training (IMT) had clinically significant improvements in dyspnea in patients with chronic obstructive pulmonary disease (COPD). Given the similar symptomatology of long COVID and COPD, the use of combined IMT is frequently observed in the studies included in this review. Other studies, as previously mentioned, also demonstrate this characteristic of combined methods developed and tested in post-COVID-19 intervention protocols [2,3,38].

It was also observed that among the training protocols with significant improvement in dyspnea and/or fatigue, the vast majority included aerobic training as one of the combined methods. Thus, aerobic exercise appears to be a fundamental part of this rehabilitation, and when combined with IMT, it emerges as a highly favorable strategy.

Targeting effective treatments for long COVID is an urgent public health priority. One of the main strengths of this review is the ability to map and analyze interventions and their outcomes across experimental and quasi-experimental studies, providing a comprehensive overview of the various physical training protocols used in individuals with long COVID. Evidence suggests that physical training can improve functional capacity, pulmonary function, and cognitive and psychological outcome such as anxiety and depression [6,7,43,44,45,46] highlighting its importance in promoting quality of life among COVID-19 survivors. Moreover, recent literature has proposed psychophysiological mechanisms that may underlie the benefits of physical training and other lifestyle-based interventions. These mechanisms include serotonergic regulation, tryptophan metabolism, and modulation of the gut-brain axis, all of which may contribute to systemic homeostasis and improved stress regulation in individuals with long COVID. This expanded perspective reinforces the value of multidisciplinary approaches, integrating physical exercise with mindfulness and nutritional strategies, in the management of post-viral syndromes [41,43].

A detailed description was provided, particularly of the training protocols used in each study (Appendix A). The entire process, from the search in major databases, following the PRISMA model, evaluating methodological quality for risk of bias, to the meta-analysis, was conducted meticulously.

Future research in the field of long COVID rehabilitation should focus on standardizing protocols, conducting long-term studies, and exploring personalized and multidisciplinary approaches. Additionally, greater emphasis on tele-rehabilitation, mechanistic studies, and understanding the variability in treatment response will help refine and optimize treatment strategies for individuals with long COVID. Age and sex differences, as well as predictions of response based on biomarkers, will also be critical areas to explore for more personalized and effective rehabilitation interventions.

The study also presents its weaknesses. Despite not being our responsibility, due to the heterogeneity of the studies presented, most had a moderate to high risk of bias, which limited the potential for a more comprehensive meta-analysis. According to the five domains used for risk of bias analysis, four studies included in this review had a low risk of bias, while the others had a high or uncertain risk of bias. The fact that these studies were conducted during the COVID-19 pandemic directly contributed to this risk of bias result, as the methodological rigor of the studies had to be adapted to be feasible during the pandemic period. There is no standardized tool for measuring fatigue and/or dyspnea in people with long COVID; the questionnaires used differ, and the lack of uniform qualification regarding fatigue levels or the circumstances under which fatigue and dyspnea were measured limits some results, reducing the body of evidence for comparing studies. Due to these limitations, only two studies allowed for meta-analysis [13,28]. Review studies point out this and other limitations, and the low evidence from studies of quality [1,38,41].

The results presented corroborate with other studies, emphasizing the importance of applying different training protocols and evaluating their effects, aiming to minimize symptoms of dyspnea and fatigue in long COVID through this diversity. Although the meta-analysis indicates that training protocols did not have an effect on reducing fatigue, it should be noted that the number of studies included in this analysis was small. It is important to highlight that most studies showed positive effects on dyspnea and fatigue when examined individually. There was significant improvement in these symptoms following physical training, particularly in protocols that included aerobic training in their combination [14,15,18,23,24,25,26,27,28,29,30,33,35].

The results of this review provide highly relevant information for healthcare professionals working in clinical practice with individuals suffering from long COVID. In the field of science, given the recent COVID-19 pandemic and the complexity of understanding the etiology of long COVID, more experimental and longitudinal studies are needed to fill gaps in the mechanisms and management of the disease. There is a need for improved study quality, including standardization of assessment tools to ensure robust data. This review focuses on identifying physical training protocols used to minimize the persistent symptoms of the disease, primarily dyspnea and fatigue. It has been possible to present pertinent data as an initial guide for treatment strategies for these symptoms, highlighting the most commonly used combinations of methods and aerobic training as a crucial part of disease management. More studies using these and other protocols are needed to build a larger body of evidence supporting our findings, facilitating increasingly personalized and effective strategies for alleviating the persistent symptoms of long COVID.

## 5. Conclusions

The most commonly used physical training protocol for rehabilitation in individuals with long COVID is the combined protocol of respiratory muscle training with strength training and aerobic training. Additionally, aerobic training was present in the methods that showed the highest efficacy in alleviating symptoms of dyspnea and/or fatigue. The meta-analysis revealed that the physical training protocol did not have a significant effect on reducing fatigue among groups of people with long COVID who received the intervention compared to the control group. However, more studies are needed for a more robust analysis. Still, the adoption of this strategy is suggested as a potential approach for alleviating these symptoms of long COVID.

## Figures and Tables

**Figure 1 healthcare-13-01897-f001:**
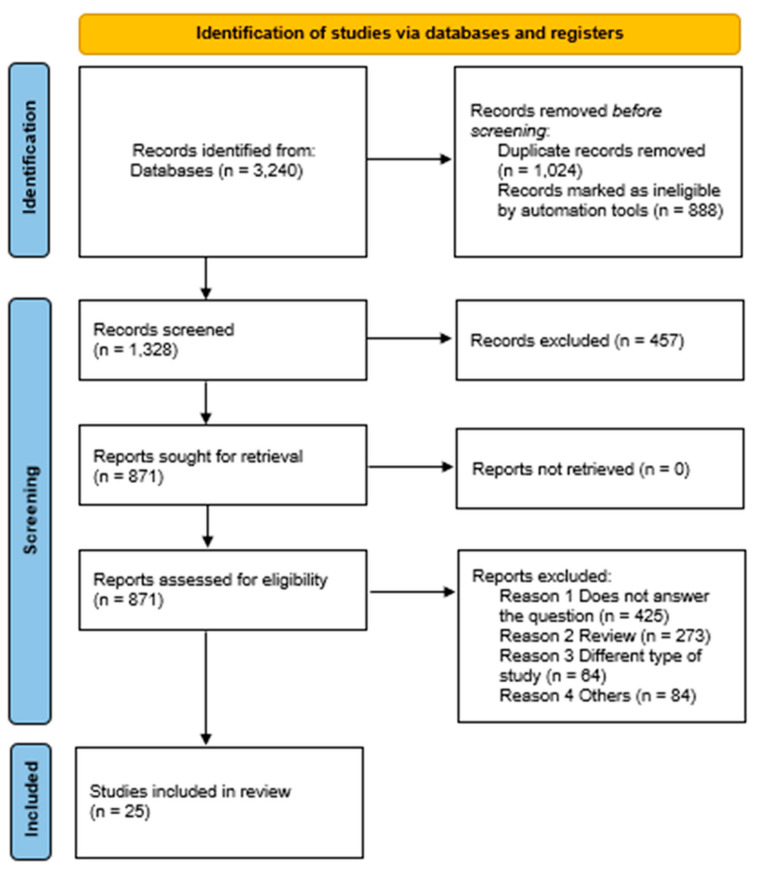
PRISMA 2020 flowchart for new systematic reviews that only included database and registry searches.

**Figure 2 healthcare-13-01897-f002:**
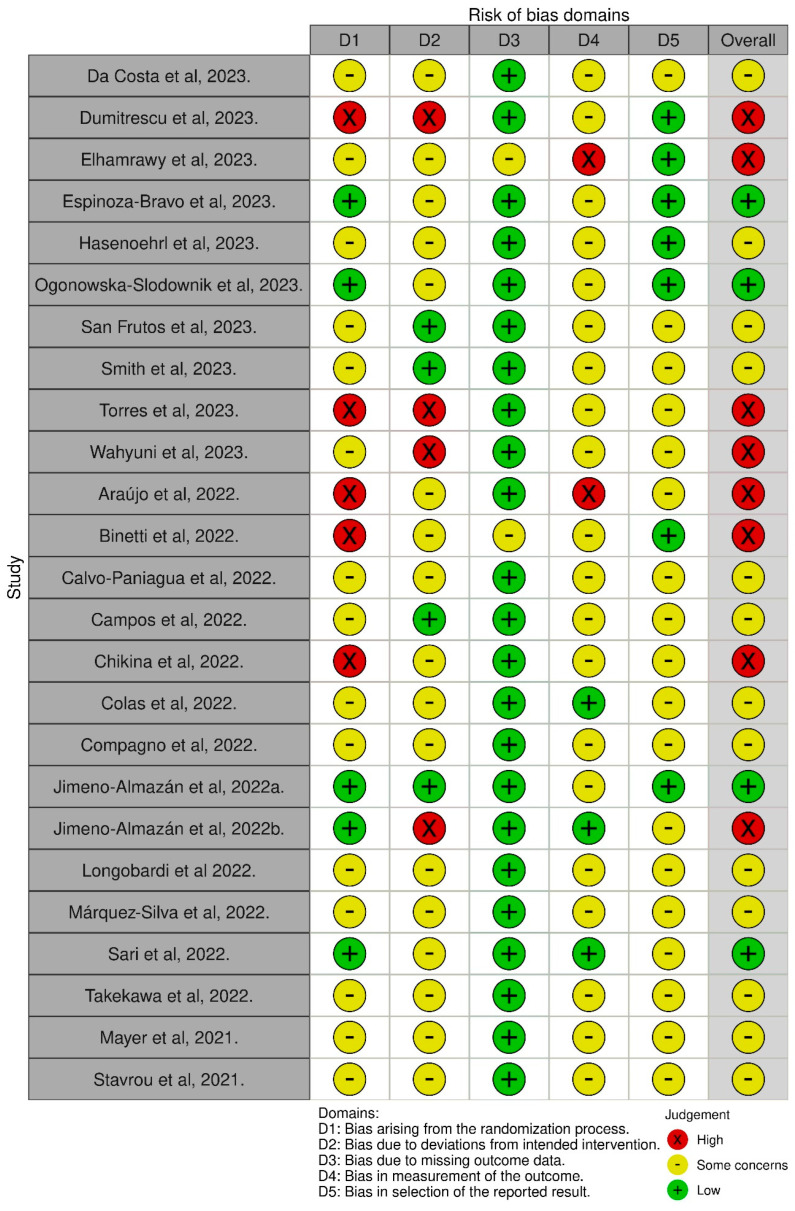
Risk of bias of included studies [11,12,13,14,15,16,17,18,19,20,21,22,23,25,26,27,28,29,30,31,32,33,34,35].

**Figure 3 healthcare-13-01897-f003:**
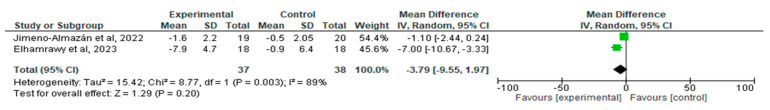
Forest plot illustrating the effect of the physical training protocol on fatigue in people with long-COVID [13,28].

**Table 1 healthcare-13-01897-t001:** Search strategy used in the databases.

Database	Search Strategy	Number of Studies Found
**PubMed**	(“Post-Acute COVID-19 Syndrome[Mesh]” OR “COVID-19 Syndrome, Post-Acute” OR “Post-Acute COVID-19 Syndromes” OR “Long Haul COVID-19” OR “COVID-19, Long Haul” OR “Long Haul COVID 19” OR “Long Haul COVID-19s” OR “Post Acute COVID-19 Syndrome” OR “Post Acute COVID 19 Syndrome” OR “Long COVID” OR “Post-Acute Sequelae of SARS-CoV-2 Infection” OR “Post Acute Sequelae of SARS-CoV 2 Infection” OR “Post-COVID Conditions” OR “Post COVID Conditions” OR “Post-COVID Condition” OR “Long-Haul COVID” OR “COVID, Long-Haul” OR “Long Haul COVID” OR “Long-Haul COVIDs”) AND (Exercise [Mesh] OR Exercises OR “Physical Activity” OR “Activities, Physical” OR “Activity, Physical” OR “Exercise, Physical” OR “Exercises, Physical” OR “Physical Exercise” OR “Physical Exercises” OR “Acute Exercise” OR “Acute Exercises” OR “Exercise, Acute” OR “Exercises, Acute” OR “Exercise, Isometric” OR “Exercises, Isometric” OR “Isometric Exercises” OR “Isometric Exercise” OR “Exercise, Aerobic” OR “Aerobic Exercise” OR “Aerobic Exercises” OR “Exercises, Aerobic” OR “Exercise Training” OR “Exercise Trainings” OR “Training, Exercise” OR “Trainings, Exercise”) AND (Fatigue [Mesh] OR Lassitude OR Dyspnea OR “Shortness of Breath” OR “Breath Shortness” OR Breathlessness OR Orthopnea OR “Recumbent Dyspnea” OR “Dyspnea, Recumbent” OR “Platypnea” OR Trepopnea OR “Rest Dyspnea” OR “Dyspnea, Rest” OR “Dyspneas, Rest”)	260
**Web of Science**	(“Post-Acute COVID-19 Syndrome” OR “COVID-19 Syndrome, Post-Acute” OR “Post-Acute COVID-19 Syndromes” OR “Long Haul COVID-19” OR “COVID-19, Long Haul” OR “Long Haul COVID 19” OR “Long Haul COVID-19s” OR “Post Acute COVID-19 Syndrome” OR “Post Acute COVID 19 Syndrome” OR “Long COVID” OR “Post-Acute Sequelae of SARS-CoV-2 Infection” OR “Post Acute Sequelae of SARS-CoV 2 Infection” OR “Post-COVID Conditions” OR “Post COVID Conditions” OR “Post-COVID Condition” OR “Long-Haul COVID” OR “COVID, Long-Haul” OR “Long Haul COVID” OR “Long-Haul COVIDs”) AND (Exercise OR Exercises OR “Physical Activity” OR “Activities, Physical” OR “Activity, Physical” OR “Exercise, Physical” OR “Exercises, Physical” OR “Physical Exercise” OR “Physical Exercises” OR “Acute Exercise” OR “Acute Exercises” OR “Exercise, Acute” OR “Exercises, Acute” OR “Exercise, Isometric” OR “Exercises, Isometric” OR “Isometric Exercises” OR “Isometric Exercise” OR “Exercise, Aerobic” OR “Aerobic Exercise” OR “Aerobic Exercises” OR “Exercises, Aerobic” OR “Exercise Training” OR “Exercise Trainings” OR “Training, Exercise” OR “Trainings, Exercise”) AND (Fatigue OR Lassitude OR Dyspnea OR “Shortness of Breath” OR “Breath Shortness” OR Breathlessness OR Orthopnea OR “Recumbent Dyspnea” OR “Dyspnea, Recumbent” OR “Platypnea” OR Trepopnea OR “Rest Dyspnea” OR “Dyspnea, Rest” OR “Dyspneas, Rest”)	219
**Embase**	(‘post-acute COVID-19 syndrome’/exp OR ‘post-acute COVID-19 syndrome’ OR ‘COVID-19 syndrome, post-acute’ OR ‘post-acute COVID-19 syndromes’ OR ‘long haul COVID-19’/exp OR ‘long haul COVID-19’ OR ‘COVID-19, long haul’ OR ‘long haul COVID 19’/exp OR ‘long haul COVID 19’ OR ‘long haul COVID-19s’ OR ‘post acute COVID-19 syndrome’/exp OR ‘post acute COVID-19 syndrome’ OR ‘post acute COVID 19 syndrome’/exp OR ‘post acute COVID 19 syndrome’ OR ‘long COVID’/exp OR ‘long COVID’ OR ‘post-acute sequelae of SARS-CoV-2 infection’/exp OR ‘post-acute sequelae of SARS-CoV-2 infection’ OR ‘post acute sequelae of SARS-CoV 2 infection’/exp OR ‘post acute sequelae of SARS-CoV 2 infection’ OR ‘post-COVID conditions’ OR ‘post COVID conditions’ OR ‘post-COVID condition’/exp OR ‘post-COVID condition’ OR ‘long-haul COVID’/exp OR ‘long-haul COVID’ OR ‘COVID, long-haul’ OR ‘long haul COVID’/exp OR ‘long haul COVID’ OR ‘long-haul COVIDS’) AND (‘exercise’/exp OR exercise OR exercises OR ‘physical activity’/exp OR ‘physical activity’ OR ‘activities, physical’ OR ‘activity, physical’/exp OR ‘activity, physical’ OR ‘exercise, physical’ OR ‘exercises, physical’ OR ‘physical exercise’/exp OR ‘physical exercise’ OR ‘physical exercises’ OR ‘acute exercise’/exp OR ‘acute exercise’ OR ‘acute exercises’ OR ‘exercise, acute’ OR ‘exercises, acute’ OR ‘exercise, isometric’/exp OR ‘exercise, isometric’ OR ‘exercises, isometric’ OR ‘isometric exercises’ OR ‘isometric exercise’/exp OR ‘isometric exercise’ OR ‘exercise, aerobic’/exp OR ‘exercise, aerobic’ OR ‘aerobic exercise’/exp OR ‘aerobic exercise’ OR ‘aerobic exercises’ OR ‘exercises, aerobic’ OR ‘exercise training’/exp OR ‘exercise training’ OR ‘exercise trainings’ OR ‘training, exercise’ OR ‘trainings, exercise’) AND (‘fatigue’/exp OR fatigue OR ‘lassitude’/exp OR lassitude OR ‘dyspnea’/exp OR dyspnea OR ‘shortness of breath’/exp OR ‘shortness of breath’ OR ‘breath shortness’ OR ‘breathlessness’/exp OR breathlessness OR ‘orthopnea’/exp OR orthopnea OR ‘recumbent dyspnea’ OR ‘dyspnea, recumbent’ OR ‘platypnea’/exp OR ‘platypnea’ OR trepopnea OR ‘rest dyspnea’ OR ‘dyspnea, rest’ OR ‘dyspneas, rest’)	904
**Scopus**	(“Post-Acute COVID-19 Syndrome” OR “COVID-19 Syndrome, Post-Acute” OR “Post-Acute COVID-19 Syndromes” OR “Long Haul COVID-19” OR “COVID-19, Long Haul” OR “Long Haul COVID 19” OR “Long Haul COVID-19s” OR “Post Acute COVID-19 Syndrome” OR “Post Acute COVID 19 Syndrome” OR “Long COVID” OR “Post-Acute Sequelae of SARS-CoV-2 Infection” OR “Post Acute Sequelae of SARS-CoV 2 Infection” OR “Post-COVID Conditions” OR “Post COVID Conditions” OR “Post-COVID Condition” OR “Long-Haul COVID” OR “COVID, Long-Haul” OR “Long Haul COVID” OR “Long-Haul COVIDs”) AND (Exercise OR Exercises OR “Physical Activity” OR “Activities, Physical” OR “Activity, Physical” OR “Exercise, Physical” OR “Exercises, Physical” OR “Physical Exercise” OR “Physical Exercises” OR “Acute Exercise” OR “Acute Exercises” OR “Exercise, Acute” OR “Exercises, Acute” OR “Exercise, Isometric” OR “Exercises, Isometric” OR “Isometric Exercises” OR “Isometric Exercise” OR “Exercise, Aerobic” OR “Aerobic Exercise” OR “Aerobic Exercises” OR “Exercises, Aerobic” OR “Exercise Training” OR “Exercise Trainings” OR “Training, Exercise” OR “Trainings, Exercise”) AND (Fatigue OR Lassitude OR Dyspnea OR “Shortness of Breath” OR “Breath Shortness” OR Breathlessness OR Orthopnea OR “Recumbent Dyspnea” OR “Dyspnea, Recumbent” OR “Platypnea” OR Trepopnea OR “Rest Dyspnea” OR “Dyspnea, Rest” OR “Dyspneas, Rest”)	473
**LILACS**	(“Post-Acute COVID-19 Syndrome” OR “COVID-19 Syndrome, Post-Acute” OR “Post-Acute COVID-19 Syndromes” OR “Long Haul COVID-19” OR “COVID-19, Long Haul” OR “Long Haul COVID 19” OR “Long Haul COVID-19s” OR “Post Acute COVID-19 Syndrome” OR “Post Acute COVID 19 Syndrome” OR “Long COVID” OR “Post-Acute Sequelae of SARS-CoV-2 Infection” OR “Post Acute Sequelae of SARS-CoV 2 Infection” OR “Post-COVID Conditions” OR “Post COVID Conditions” OR “Post-COVID Condition” OR “Long-Haul COVID” OR “COVID, Long-Haul” OR “Long Haul COVID” OR “Long-Haul COVIDs”) AND (Exercise OR Exercises OR “Physical Activity” OR “Activities, Physical” OR “Activity, Physical” OR “Exercise, Physical” OR “Exercises, Physical” OR “Physical Exercise” OR “Physical Exercises” OR “Acute Exercise” OR “Acute Exercises” OR “Exercise, Acute” OR “Exercises, Acute” OR “Exercise, Isometric” OR “Exercises, Isometric” OR “Isometric Exercises” OR “Isometric Exercise” OR “Exercise, Aerobic” OR “Aerobic Exercise” OR “Aerobic Exercises” OR “Exercises, Aerobic” OR “Exercise Training” OR “Exercise Trainings” OR “Training, Exercise” OR “Trainings, Exercise”) AND (Fatigue OR Lassitude OR Dyspnea OR “Shortness of Breath” OR “Breath Shortness” OR Breathlessness OR Orthopnea OR “Recumbent Dyspnea” OR “Dyspnea, Recumbent” OR “Platypnea” OR Trepopnea OR “Rest Dyspnea” OR “Dyspnea, Rest” OR “Dyspneas, Rest”)	222(2)
**EBSCO ***	(“Post-Acute COVID-19 Syndrome” OR “COVID-19 Syndrome, Post-Acute” OR “Post-Acute COVID-19 Syndromes” OR “Long Haul COVID-19” OR “COVID-19, Long Haul” OR “Long Haul COVID 19” OR “Long Haul COVID-19s” OR “Post Acute COVID-19 Syndrome” OR “Post Acute COVID 19 Syndrome” OR “Long COVID” OR “Post-Acute Sequelae of SARS-CoV-2 Infection” OR “Post Acute Sequelae of SARS-CoV 2 Infection” OR “Post-COVID Conditions” OR “Post COVID Conditions” OR “Post-COVID Condition” OR “Long-Haul COVID” OR “COVID, Long-Haul” OR “Long Haul COVID” OR “Long-Haul COVIDs”) AND (Exercise OR Exercises OR “Physical Activity” OR “Activities, Physical” OR “Activity, Physical” OR “Exercise, Physical” OR “Exercises, Physical” OR “Physical Exercise” OR “Physical Exercises” OR “Acute Exercise” OR “Acute Exercises” OR “Exercise, Acute” OR “Exercises, Acute” OR “Exercise, Isometric” OR “Exercises, Isometric” OR “Isometric Exercises” OR “Isometric Exercise” OR “Exercise, Aerobic” OR “Aerobic Exercise” OR “Aerobic Exercises” OR “Exercises, Aerobic” OR “Exercise Training” OR “Exercise Trainings” OR “Training, Exercise” OR “Trainings, Exercise”) AND (Fatigue OR Lassitude OR Dyspnea OR “Shortness of Breath” OR “Breath Shortness” OR Breathlessness OR Orthopnea OR “Recumbent Dyspnea” OR “Dyspnea, Recumbent” OR “Platypnea” OR Trepopnea OR “Rest Dyspnea” OR “Dyspnea, Rest” OR “Dyspneas, Rest”)	478
**Google scholar**	(Post-Acute COVID-19 Syndrome) AND (exercise) AND (fatigue OR dyspnea)	100
**medRxiv**	(Post-Acute COVID-19 Syndrome) AND (exercise) AND (fatigue OR dyspnea)	584

*** The articles found in EBSCO were those retrieved from the following databases:** CINAHL with Full Text 63; MEDLINE Complete 219; Academic Search Premier 119; SPORTDiscus with Full Text 22; Food Science Source 17.

**Table 2 healthcare-13-01897-t002:** Characteristics of the studies, participants, assessment instruments, training protocols, interventions and stages of the systematic review.

References	Objective	Types of Intervention	Effects on Fatigue	Effects on Dyspnea
Vieira Da Costa et al., 2023. Brazil [11]	To evaluate the effectiveness of a rehabilitation protocol on pulmonary and respiratory muscles and the thickness of the quadriceps femoris and diaphragm in patients with post-COVID-19 syndrome.	RMT + Strength Training + Aerobic 6 weeks. RMT: 3 × 10 at 40% inspiratory pressure load; Strength training 2× per week; Anterior Elevation 3 × 1 min pause 1 min. Aerobic: Treadmill at 60–70% and 70–80% of max HR.	↔	NA
Dumitrescu et al., 2023. Romania [12]	To assess the impact of COVID-19 on cardiopulmonary health and the effectiveness of various rehabilitation interventions. This study hypothesizes that post-COVID-19 patients present distinct characteristics and recovery dynamics that significantly influence their response to specific rehabilitation interventions.	Aerobic + RMT + Strength Training 3 months. Aerobic: starting with a bike at 40–50% max effort 3× per week, gradually adding walking; frequency increased to 5× and intensity varied based on individual responses, such as HIIT application. RMT: guided coughing exercises and drainage posture (airway clearance and mucus removal). Strength training: 2× per week with 1–2 kg weights and low elasticity resistance bands (increasing gradually to 3–4× per week). Flexibility 2–3× per week.	NA	↔
Elhamrawy et al., 2023. Egypt [13]	To understand the impact of Tai Chi versus aerobic training on fatigue and functional performance in elderly post-COVID.	Tai Chi (TC) or Aerobic 3 months. TC: 4× per week for 60 min; 10′ warm-up; 40′ main session; 10′ relaxation: 7 TC movements: controlled weight shifting, ankle swings, and forward-backward and lateral steps (4–6 Borg Scale). Aerobic: 4× per week for 60 min; warm-up: 10′ static stretches targeting the trunk and limbs; 20′ general muscle strengthening with 5 kg; Treadmill walking for 15–20 min at moderate effort of 12–26 km/h; 10′ relaxation.	↑ ↑	NA
Espinoza-Bravo et al., 2023. Spain [14]	To compare the clinical effects of two tele-rehabilitation programs on long COVID symptoms.	Functional + RMT or Aerobic + RMT 3× per week for 8 weeks. App: https://fisiotrack.com/ (accessed on 5 January 2025) Functional: Squats, lateral squats, hip lifts, bench press, and rowing; 2–3× 10 reps. 1′ pause; 4–6 exercises, increasing difficulty every 2 weeks. Aerobic: progressive walking with weekly load adjustments from 25′ to 45′ (25′ in week 1 to 45′ in week 8). Walking pace allowed maintaining a fluid conversation. RMT: diaphragmatic and pursed-lip breathing 5× per week, 3 × 15 reps.	↑ ↑	NA
Hasenoehrl et al., 2023. Áustria [15]	To assess the effects of physical exercise on post-COVID-19 symptoms on physical/mental abilities and work capacity within a workplace health promotion project among healthcare workers.	Strength Training + Aerobic Strength training: 2× training sessions over 8 weeks supervised, circuit training with body weight and resistance bands, including 8 full-body exercises. Squats, glute bridges, hip walks, back extensions at 45°, push-ups, low row, planks, and shoulder external rotations with varying difficulty levels based on fitness. Starting with 30″ exercise and 30″ pause, progressing every 2 weeks by adding 10″ until reaching 1′/1′. Initial PSE 7–8 and from week 3 PSE 9–10. Aerobic: unsupervised aerobic exercises—recommendations: at least 3× 20′ of moderate exercise per week at LV1. Note: Severe cases showed better levels of improvement compared to mild cases. Note 2: PCFS corrected by Bonferroni method shows strong association between relative VO2 peak, 30-s STS, and 6MWT.	↑	NA
Ogonowska-Slodownik et al., 2023. Poland [16]	To analyze the effectiveness of water- and land-based training programs on exercise capacity, fatigue, and secondary outcomes on health-related quality of life in children with post-COVID-19 condition.	Aerobic or Aquatic 2× for 8 weeks, 45′. Warm-up: 8′ and relaxation at the end for 5′. Aquatic: Punch, kick, stationary running, and breath control. Aerobic: 2 circuit stations, each with 5 stations with different resistance exercises for upper and lower limbs, 1′ exercise and 15″ pause. Intensity 6–8 on the Pictorial scale. Aquatic and land exercises were combined as closely as possible in terms of intensity, duration, and muscle groups trained. (No significant differences were found in total fatigue level and individual fatigue symptoms).	↔ ↔	NA
De La Plaza San Frutos et al., 2023. Spain [17]	To evaluate the outcomes of a tele-respiratory rehabilitation program in post-COVID-19 patients ranging from mild to critical COVID-19 without vaccination.	RMT + Stretching RMT: 10 online sessions with diaphragmatic breathing: costal expansion, flexion, and abduction of upper limbs; active respiratory cycle; yoga-based breathing exercises and progressive muscle relaxation. Stretching: specific for neck and thoracic muscles.	NA	↑
Smith et al., 2023. Europe, África e Ásia [18]	To evaluate the clinical efficacy of a new 12-week combined community rehabilitation program for individuals with long COVID.	Programa Nuffield Health: www.nuffieldhealth.com/covid-rehab (accessed on 5 January 2025) Nuffield Health Program: Participants used a diary to record exercises and progress. Exercises: 12 weeks (6 weeks remote and 6 weeks in-person in a community setting), 3 × 45 min per week (36 total) including a combination of cardiovascular, strength, and mobility exercises. Exercise intensity, volume, movement complexity, range of motion, and stability were prescribed according to participants’ functional capacity and fitness. Weekly distribution: Week 1 cardio; Week 7 cardio; Week 2 mobility; Week 8 mobility; Week 3 strength; Week 9 strength; Week 4 mobility; Week 10 mobility; Week 5 cardio; Week 11 strength; Week 6 strength; Week 12 cardio.	NA	↑
Torres et al., 2023. South Africa [19]	To help standardize exercise for post-COVID (applying the CERT by describing the 16 essential items).	Multicomponent Aerobic: 3× per week for 7–10′. Progression: increase by 1–2′ each week and PSE 6–11 (gradually). Strength Training: 1× 4–8 reps on equipment (add 2 reps every week, and when reaching 10 reps, add 1 exercise and 1 set each week until 3 × 10 reps). Balance exercises; flexibility exercises (2–4 exercises based on HR, dyspnea level, technique, symptoms for sedentary/beginners).	↔	↔
Wahyuni et al., 2023. Indonesia [20]	To analyze the effect of educational videos on therapeutic exercises on the functional status of subjects after hospitalization for COVID-19.	Strength Training Basic Strength and Balance Exercises: 2–3 times per day with 5–10 reps for each movement. Phase 1: seated: Strengthening and balance. Phase 2: Squat (sit-to-stand and gait alignment or assistance).	↔	NA
Araújo et al., 2022. Brazil [21]	To verify the effects of a cardiopulmonary rehabilitation program consisting of continuous moderate-intensity aerobic and resistance training on pulmonary function, respiratory muscle strength, maximal and submaximal exercise tolerance, fatigue, and quality of life in post-COVID-19 patients.	Aerobic + Strength Training Aerobic: 40′ on treadmill, 5′ warm-up; 30′ conditioning; 5′ relaxation. Strength Training: 2× per week, 3× 8–12 reps. Biceps, triceps, shoulder abductors; quadriceps, hip abductors, and calf muscles. 60% of 1RM with progression at the 6th session. A total of 12 sessions.	↔	NA
Binetti et al., 2022. Spain [22]	To analyze the impact of a supervised exercise program on the clinical evolution of long COVID patients with persistent fatigue as the main symptom and to identify whether specific blood biomarkers could predict rehabilitation response in a cohort from a high-prevalence area.	Aerobic + Strength Training (Upper Limbs) 3 months with 12–20 supervised physiotherapy sessions; Warm-up: 5–10′ stretching and light exercises; Main: aerobic on bike starting at 10′ progressing to 30′. Intensity according to effort and tolerance based on HR. 1× per week upper limb muscle strengthening exercises.	↔	↔
Calvo-Paniagua et al., 2022. Spain [23]	To analyze whether a tele-rehabilitation program can improve self-perception of physical exertion in patients with post-COVID fatigue and dyspnea.	Aerobic + Active Mobilizations and Motor Control + Respiratory Muscle Training (RMT) RMT: airway clearance exercises and respiratory exercises (in all sessions). 3 months with 18 sessions, 3× per week, 40′ on alternate days via videoconference (Zoom). Session 1: Postural Ergonomics; Session 2–5 RMT: diaphragmatic breathing, costal breathing, pursed-lip breathing, and airway clearance; Session 6–8: Physical conditioning with increasing intensity: active mobilizations of the cervical, dorsal, and lumbar spine, active mobilizations of lower and upper limbs + core training with motor control exercises; Session 9: Balance + Dynamic control seated exercises + gait exercises; Session 10–11: functional exercises and plyometric exercises + occupational therapy for daily activities; Session 12–18: functional exercises + occupational therapy + aerobic training: walking at a tolerable speed.	NA	↑
Campos et al., 2022. Brazil [24]	To evaluate the effects of an 8-week in-person rehabilitation program for COVID-19 on fatigue and dyspnea, exercise capacity, pulmonary function, cognitive function, anxiety and depression symptoms, and peripheral muscle strength compared to a remote monitoring group.	RMT + Aerobic + Strength Training + Stretching 8 weeks, 2× per week, 80′. RMT: respiratory exercises, energy conservation techniques. Aerobic: Treadmill at 75% Borg speed of 4–6. Warm-up: 5′ and 30′ at target intensity (moderate). Strength Training: 80% of 1RM with 3 × 10 reps, 1–2′ pause for upper and lower limbs. Stretching: muscles trained were stretched.	↑	↑
Chikina et al., 2022. Russia [25]	To investigate the effectiveness of physical rehabilitation in treating post-COVID syndrome in patients with lung injuries caused by coronavirus infection.	RMT + Strength Training + Aerobic RMT: Loaded breathing and positive expiratory pressure breathing (respiratory simulators). Strength Training: Exercises with a gym stick, elastic bands, and 0.5–2 kg dumbbells. Aerobic: Stepper (walking simulator).	NA	↑
Colas et al., 2022. France [26]	To evaluate fatigue in patients with prolonged symptoms after COVID-19 infection who received a mixed program of adapted physical activity remotely and therapeutic education. The secondary aim was to assess the efficacy and safety of this training method through aerobic and anaerobic parameters.	Aerobic + Strength Training or RMT Aerobic: 12 sessions; 3× per week; 45′ with the first week in-person and subsequent sessions online (training at ventilatory threshold 1 (VT1) with progression to VT2 on a cycle ergometer from week 4 interval training. Strength Training: 15′ full-body circuit with body weight, with light to moderate intensity according to the Borg scale. Note: Exercise was stopped if the heart rate was >80% max HR (vigorous exercise) and/or if the perceived effort was >6/10 on the Borg scale. RMT: Traditional community physical rehabilitation 3× per week for 4 weeks.	↑ ↑	NA
Compagno et al., 2022. Italy [27]	To evaluate the efficacy, safety, and feasibility of an out-of-hospital multidisciplinary rehabilitation program (MDR) based on physical and psychological reconditioning to reduce symptoms and improve fitness and psychological parameters in long COVID patients.	ACSM Program + RMT based on meditation Training 3× per week for 90′ (10′ mixed warm-up + 45′ main session (strength training + 35′ aerobic and 5′ relaxation). RMT: muscle relaxation techniques, breath control, and guided relaxation. Aerobic: cycle ergometer and continuous treadmill moderate (60–80% VO2 peak). Strength Training: performed with variable loads of 30 to 50% of 1RM (pectoral, lumbar, leg press, leg extension, adductor machine, and deltoids). Stretching: at the end as relaxation.	↑	↑
Jimeno-Almazán et al., 2022 Spain [28]	Determine the effectiveness of physical exercise, respiratory muscle training, and the World Health Organization (WHO) self-management recommendation leaflet on recovery of physical fitness, quality of life, and symptoms.	Concurrent Training (CT)—Strength Training + Aerobic or RMT or Concurrent Training + RMT or WHO Guidelines CT: 8 weeks, 3 sessions per week, 2× Strength training: 3 × 8 repetitions, 4 exercises squats, bench press, deadlifts and bench pulls. 50% RM constant programming model—intra-set intensity and volume kept constant throughout the training plan. Strength training was combined with moderate-intensity variable aerobic training (MIVT: 4–6 × 3–5 min at 70–80% heart rate reserve [HRR]/2–3 min at 55–65% of FCR) TMR: Inspiratory muscle training with PowerBreath Classic Heath Series Mechanical Threshold Devices 1 set of 30 repetitions [62.5 ± 4.6% of IMP (Maximum Inspiratory Pressure)], preceded by a warm-up set twice daily, all the days of the week TC + TMR: the combination of the previous protocols. WHO: Controlled breathing, light walking, stretching and balance exercises (Borg 0–1). Phase 2: Add light daily household chores (Borg 2–3). Phase 3: brisk walking, going up and down stairs and running for up to 30′ adding 15′ strength exercises (Borg 4–5). Phase 4: moderate intensity exercises with coordination and skill in running, cycling, swimming and dancing (Borg 5–7). Phase 5: the participant returned to their usual routine (Borg 8–10).	↑ ↔ ↑ ↔	↑ ↔ ↑ ↓
Jimeno-Almazán et al., 2022 Spain [29]	Compare the outcomes of patients with post-COVID-19 condition undergoing supervised therapeutic exercise intervention or following the WHO (World Health Organization) self-management rehabilitation leaflet.	Concurrent Training (CT)—Strength Training + Aerobic or WHO Guidelines post-COVID CT: 8 weeks, 3 sessions per week, 2× Strength training: 3 × 8 repetitions, 4 exercises squats, bench press, deadlifts and bench pulls. 50% RM constant programming model—intra-set intensity and volume kept constant throughout the training plan. Strength training was combined with moderate-intensity variable aerobic training (MIVT: 4–6 × 3–5 min at 70–80% heart rate reserve [HRR]/2–3 min at 55–65% of FCR). Isolated aerobic: 1 day of continuous light intensity training (LICT: 30–60min, 65–70% HRR) WHO: 30′, 5× week at an intensity that allows breathless speech plus strength exercises in 3 weekly sessions (3 × 10 repetitions of the 7 recommended exercises). Note: adapted and supervised multicomponent exercise program from the ACSM guidelines for chronic obstructive pulmonary disease and cardiovascular disease.	↔ ↔	↑ ↔
Longobardi et al. 2022. Brazil [30]	Recovery of general physical condition in post-COVID patients.	Aerobic + Strength Training + Stretching—HBET: home training 3× a week, aerobic exercises for strengthening and flexibility. 1 supervised session (online) and 2 unsupervised sessions. Aerobic: 2 sets of 10 min/day of walking at “very light” to “reasonably light” intensity (Borg scale 9–11) following week, progression with a single 45-min walking session at “somewhat difficult” to “strong” (Borg scale 14–16) Strength training: 6 strengthening exercises of 3 to 4 sets per exercise of 10 to 15 repetitions, with a self-suggested recovery interval between sets. Stretching: Active stretching exercises for major muscle groups were prescribed as relaxation.	↑	↑
Márquez-Silva et al., 2022. Mexico [31]	Generate an exercise program to improve the physical condition of patients with the syndrome Post-COVID-19.	Strength training + Aerobic + Balance + RMT 10 sessions 4× per week, 60–90′ per session based on RPE. The session was divided into: warm-up; central phase where strength and muscular endurance exercises were worked on; aerobic training; balance exercises and breathing exercises. Strength training: bench press; military bench press; squat; deadlift. Aerobic: walking. Balance: - TMR: - Note: Own preparation.	↔	↔
Sari et al., 2022. Turkey [32]	To investigate the effectiveness of inspiratory muscle training on exercise capacity, lower muscle strength, dyspnea, anxiety-depression, quality of life, physical activity and fatigue in coronavirus disease 2019 (COVID-19) patients with respiratory diseases.	Strength training + RMT or RMT (inspiratory) TMR: diaphragmatic breathing; chest expansion; exercises to increase chest compliance with respiratory control with an exercise band. 5–10 repetitions of 5 to 10 min, 3 sets/day. Strength training: squats and clinical bridge exercises (based on pilates), for 6 weeks every day 3 × 10 repetitions/day. TMR (inspiratory): 30 min, 3 times, 7 days for 6 weeks, deep diaphragmatic breathing for 8 to 10 breaths. Loading resistance was increased every week.	↑ ↑	↑ ↔
Takekawa et al., 2022. Japan [33]	Determine the impact of Unsupervised Pulmonary Rehabilitation. We describe here a nephrectomized patient with severe COVID-19 infection who required extracorporeal membrane oxygenation (ECMO) during admission to the intensive care unit (ICU), but made a full recovery and returned to society after rehabilitation therapy.	RMT + Strength training + Aerobic 3× a day, 30′ total with 3 sets each time, 6 days a week. Self-training focused on strengthening the muscles of the trunk, upper and lower limbs and expiratory muscles. TMR: trunk stretching, shoulder girdle and neck exercises and deep breathing. Strength training: joint range of motion and muscle strengthening exercises. Non-equipment-based resistance training. 8–12 reps. Aerobic: in “easy” PSE; walking exercise and going up and down stairs.	↑	NA
Mayer et al., 2021. USA [34]	Provide the clinical presentation and physical therapy management of a patient with post-COVID syndrome.	Aerobic + Strength training + RMT Fortnightly sessions for 8 weeks totaling 16 training sessions (15 sessions were held) from 40–80′. The first 4 were supervised with education to be carried out at home. Aerobic: Upper limb exercise bike, treadmill, running and dancing. 60–80% of HR max. in test with at least 15′. Strength training: ‘10–20’ of 10–15 repetitions focused on multi-joint or compound exercises. Based on PSEm when <4/10 with increased load and/or number of repetitions. Start with dumbbells (1.3kg) and leg weights sitting and standing. Progressing to functional movements with squats and weight gain. TMR: controlled diaphragmatic breathing techniques with relaxation and mindfulness. Diaphragmatic breathing combined with general core and trunk exercises (sitting lumbar extension and flexion with core activation, cat-cow exercise, child’s pose and bird-dog exercises). Note: post-exertion malaise.	↓	↓
Stavrou et al., 2021. Greece [35]	Determine the impact of Unsupervised Pulmonary Rehabilitation (uns-PR) on patients recovering from COVID-19, and determine their anthropometric and biological characteristics.	Multicomponent + Breathing Exercises based on Yoga 8 weeks, 3 weekly sessions, around 100 min. Flexibility and mobility: Warm-up/recovery: Child’s Pose-Prayer Stretch, Doorway Stretch, Quadriceps Stretch. Aerobic: walking on a hard, flat surface marking the distance covered (50′). Yoga: for breathing and proprioception (20′). Strength training: multi-joint exercises; lateral raise with dumbbells; squats with dumbbells, chair lunges, seated leg raises, elbow flexion-extension on the chest with medicine ball.	↔	↑

Note: ↑ = significant improvement; ↓ = worsened; ↔ = no change; NA = not assessed. RMT: respiratory muscle training; TC: Tai Chi; CT: concurrent training; UL: upper limbs; ACSM: American College of Sports Medicine; mRPE: modified Rating of Perceived Exertion scale.

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
