# Peer review of "Physical Training Protocols for Improving Dyspnea and Fatigue in Long COVID: A Systematic Review with Meta-Analysis"

_healthcare, 2025, doi:10.3390/healthcare13151897_

Round 1
Reviewer 1 Report
Comments and Suggestions for Authors
This is a review and meta-analysis of studies that evaluated the physical training protocols that were used for people with long COVID to alleviate symptoms of dyspnea and fatigue. Several physical training protocols that used a combination of different techniques were identified the most common were respiratory muscle training with strength training, and aerobic training. There is a great heterogeneity among the studies, they used different methods of assesing dispnea and fatigue. Although the meta-analysis did not find a significant effect on fatigue, most individual studies showed improvements in both symptoms, especially with combined methods.
- The main question of the study is what physical training protocols are used to improve dyspnea and fatigue in individuals with long COVID, and how effective they are
- The topic is highly relevant and addresses a gap, as there is limited evidence on rehabilitation strategies for long COVID. This review helps guide future research and clinical practice.
- It systematically summarizes different exercise protocols, compares their outcomes, highlights the lack of standardized approaches, and evaluates study quality. This offers a clearer foundation than individual or narrative reviews.
- The conclusions reflect the evidence and acknowledge limitations, appropriately answering the main question.
Overall, nice presentation of the strengnths and limitations of the studies. Minor suggestions:
- it might be helpful to make Tables 1 and 2 more consise.
- the authors can include more studies in the meta-analysis as only 2 were included. Also, they could do a subgroup analyses so as to explain the variations in different studies and maybe identify which patients could benefit the most. For example they could include groups based on symptom severity (mild vs moderate) or age groups
- The references are up-to-date and relevant. Some references that could be included are the following:
Author Response
Reviewer 1
Response:
We sincerely thank Reviewer 1 for the thorough and constructive feedback provided. Your thoughtful comments and suggestions greatly contributed to improving the clarity, depth, and overall quality of our manuscript. We carefully considered each point raised and have revised the manuscript accordingly. We hope the changes made address your concerns and meet your expectations.
Open Review
(x) I would not like to sign my review report
( ) I would like to sign my review report
Quality of English Language
|
( ) The English could be improved to more clearly express the research. |
|
Yes |
Can be improved |
Must be improved |
Not applicable |
|
|
Does the introduction provide sufficient background and include all relevant references? |
(x) |
( ) |
( ) |
( ) |
|
Is the research design appropriate? |
(x) |
( ) |
( ) |
( ) |
|
Are the methods adequately described? |
(x) |
( ) |
( ) |
( ) |
|
Are the results clearly presented? |
(x) |
( ) |
( ) |
( ) |
|
Are the conclusions supported by the results? |
(x) |
( ) |
( ) |
( ) |
|
Are all figures and tables clear and well-presented? |
(x) |
( ) |
( ) |
( ) |
Comments and Suggestions for Authors
This is a review and meta-analysis of studies that evaluated the physical training protocols that were used for people with long COVID to alleviate symptoms of dyspnea and fatigue. Several physical training protocols that used a combination of different techniques were identified the most common were respiratory muscle training with strength training, and aerobic training. There is a great heterogeneity among the studies, they used different methods of assesing dispnea and fatigue. Although the meta-analysis did not find a significant effect on fatigue, most individual studies showed improvements in both symptoms, especially with combined methods.
- The main question of the study is what physical training protocols are used to improve dyspnea and fatigue in individuals with long COVID, and how effective they are
- The topic is highly relevant and addresses a gap, as there is limited evidence on rehabilitation strategies for long COVID. This review helps guide future research and clinical practice.
- It systematically summarizes different exercise protocols, compares their outcomes, highlights the lack of standardized approaches, and evaluates study quality. This offers a clearer foundation than individual or narrative reviews.
- The conclusions reflect the evidence and acknowledge limitations, appropriately answering the main question.
Overall, nice presentation of the strengnths and limitations of the studies. Minor suggestions:
- it might be helpful to make Tables 1 and 2 more consise.
- the authors can include more studies in the meta-analysis as only 2 were included. Also, they could do a subgroup analyses so as to explain the variations in different studies and maybe identify which patients could benefit the most. For example they could include groups based on symptom severity (mild vs moderate) or age groups
Response:
We appreciate the reviewer’s thoughtful suggestion. We agree that subgroup analyses could provide valuable insights. However, we did not perform them due to the high methodological heterogeneity among the included studies. The studies varied considerably in terms of population characteristics (e.g., wide age ranges, different levels of clinical severity), types and durations of interventions, and outcome measures. Furthermore, although additional studies were identified, most did not provide compatible quantitative data (e.g., means and standard deviations for the same outcomes), which precluded their inclusion in the meta-analysis. Consequently, we were limited to a narrative synthesis and highlighted these methodological limitations in the discussion section of the manuscript.
- The references are up-to-date and relevant. Some references that could be included are the following:
Clinical management of COVID-19: Living guideline [Internet]. Geneva: World Health Organization; 2022 Jun 23–. PMID: 35917394. COVID-19 rapid guideline: managing the long-term effects of COVID-19. London: National Institute for Health and Care Excellence (NICE); 2024 Jan 25. PMID: 33555768.
Response:
We thank the reviewer for the valuable suggestion. The recommended references were indeed relevant and have now been included in the revised version of the manuscript to strengthen the contextual foundation of our work.
Reviewer 2 Report
Comments and Suggestions for Authors
In this manuscript, the authors presented results of their systematic review aimed at evaluating physical training protocols for alleviating long COVID symptoms, especially dyspnea and fatigue, through a systematic review with meta-analysis.
The article is well prepared and provides readers with clinically relevant information; however, I have some recommendations to raise.
Firstly, I recommend updating the abstract with the general and specific background.
Secondly, I believe that the information presented in the column entitled "Types of intervention" at Table 2 needs to be separated into two columns: type of intervention and how this intervention was performed. Please provide in the Table how Fatigue and Dyspnea were assessed.
Next, please provide the legend of Table 2 in English in Portuguese.
Author Response
Reviewer 2
Response:
We sincerely thank Reviewer 2 for the thorough and constructive feedback provided. Your thoughtful comments and suggestions greatly contributed to improving the clarity, depth, and overall quality of our manuscript. We carefully considered each point raised and have revised the manuscript accordingly. We hope the changes made address your concerns and meet your expectations.
Open Review
( ) I would not like to sign my review report
(x) I would like to sign my review report
Quality of English Language
(x) The English could be improved to more clearly express the research.
( ) The English is fine and does not require any improvement.
|
Yes |
Can be improved |
Must be improved |
Not applicable |
|
|
Does the introduction provide sufficient background and include all relevant references? |
(x) |
( ) |
( ) |
( ) |
|
Is the research design appropriate? |
(x) |
( ) |
( ) |
( ) |
|
Are the methods adequately described? |
(x) |
( ) |
( ) |
( ) |
|
Are the results clearly presented? |
(x) |
( ) |
( ) |
( ) |
|
Are the conclusions supported by the results? |
(x) |
( ) |
( ) |
( ) |
|
Are all figures and tables clear and well-presented? |
( ) |
(x) |
( ) |
( ) |
Comments and Suggestions for Authors
In this manuscript, the authors presented results of their systematic review aimed at evaluating physical training protocols for alleviating long COVID symptoms, especially dyspnea and fatigue, through a systematic review with meta-analysis.
The article is well prepared and provides readers with clinically relevant information; however, I have some recommendations to raise.
Firstly, I recommend updating the abstract with the general and specific background.
Secondly, I believe that the information presented in the column entitled "Types of intervention" at Table 2 needs to be separated into two columns: type of intervention and how this intervention was performed. Please provide in the Table how Fatigue and Dyspnea were assessed.
Next, please provide the legend of Table 2 in English in Portuguese.
Response:
We thank the reviewer for the constructive comments. Initially, Table 2 was structured to include both the type of intervention and its implementation details in a single column to ensure a concise presentation. However, not all studies provided complete information on these aspects. Due to the heterogeneity among the studies and to facilitate a clearer understanding, we opted to keep this combined format.
Nevertheless, we have revised the table to clarify the intervention procedures wherever possible and have now added information on how fatigue and dyspnea were assessed in each study. Additionally, we have included the table legend in both English and Portuguese, as recommended.
Reviewer 3 Report
Comments and Suggestions for Authors
Dear Authors,
We appreciate the opportunity to review your manuscript, which addresses an important and timely topic regarding rehabilitation strategies for individuals experiencing long COVID symptoms. Your work contributes to the growing body of literature exploring physical training as a potential intervention for dyspnea and fatigue in this population. However, the manuscript requires substantial improvements in several areas. I hope the following comments will offer constructive suggestions to enhance its scientific rigor, clarity, and relevance to both clinical practice and research.
Introduction
The introduction omits a discussion of biopsychosocial models or mechanisms (e.g., neuroinflammation, mitochondrial dysfunction) that may underlie post-viral fatigue and dyspnea.
In addition, the term “physical training” is not clearly defined early in the text cardio, strength, neuromotor, and respiratory interventions are treated as if they were interchangeable.
Methods
What does Table 1 add compared to the flowchart? Since the search strings used are the same for all databases, repeating them, as well as the number of selected papers, may not be necessary. Also, why was a different search string used for medRxiv and Google Scholar? Consider removing the table if it does not provide additional value.
Flowchart: some words appear to be cut off in the figure, such as “identification” and “by.” Please fix these formatting issues.
The term “automation tool” should be explained more clearly in the Methods section. what tool was used, and how?
The legend in Table 2 is not in English—please translate it.
The OSF registration is a positive indication of transparency. However, combining studies with heterogeneous designs into a single meta-analytic model may not be methodologically appropriate. Only two studies were included in the meta-analysis, which is statistically and clinically insufficient to conduct a meaningful pooled analysis.
Results
The results vary widely across interventions. The lack of detailed subgroup analyses (e.g., based on long COVID severity, age, baseline fatigue levels) limits the interpretability of the findings.
Discussion
Please avoid repeating the results in the Discussion session (e.g., lines 95–99).
The discussion does not explore the underlying mechanisms by which physical training contributes to disease management; it merely states that exercise plays a key role (e.g., line 111).
Similarly, in lines 124–126, the text refers to a comprehensive, multidisciplinary treatment approach without discussing the mechanisms of action. Emerging evidence suggests that a multidisciplinary, lifestyle-based approach, combining physical activity, mindfulness practices, and nutritional interventions, can offer broader benefits in post-viral syndromes such as long COVID. For instance, Baldassano et al. (2023) introduce the Mindfulness–Exercise–Nutrition (MEN) model and discuss psychophysiological pathways (e.g., serotonin, tryptophan metabolism, and the gut–brain axis) involved in managing stress and promoting systemic homeostasis during the COVID-19 pandemic.
The discussion would benefit from a deeper literature review to better connect the proposed intervention protocol with its underlying mechanisms.
Baldassano, S.; Alioto, A.; Amato, A.; Rossi, C.; Messina, G.; Bruno, M.R.; Stallone, R.; Proia, P. Fighting the Consequences of the COVID-19 Pandemic: Mindfulness, Exercise, and Nutrition Practices to Reduce Eating Disorders and Promote Sustainability. Sustainability 2023, 15, 2120. https://doi.org/10.3390/su15032120
Author Response
Reviewer 3
Response:
We sincerely thank Reviewer 3 for the thorough and constructive feedback provided. Your thoughtful comments and suggestions greatly contributed to improving the clarity, depth, and overall quality of our manuscript. We carefully considered each point raised and have revised the manuscript accordingly. We hope the changes made address your concerns and meet your expectations.
Open Review
(x) I would not like to sign my review report
( ) I would like to sign my review report
Quality of English Language
|
( ) The English could be improved to more clearly express the research. |
|
Yes |
Can be improved |
Must be improved |
Not applicable |
|
|
Does the introduction provide sufficient background and include all relevant references? |
( ) |
(x) |
( ) |
( ) |
|
Is the research design appropriate? |
(x) |
( ) |
( ) |
( ) |
|
Are the methods adequately described? |
( ) |
(x) |
( ) |
( ) |
|
Are the results clearly presented? |
( ) |
(x) |
( ) |
( ) |
|
Are the conclusions supported by the results? |
( ) |
(x) |
( ) |
( ) |
|
Are all figures and tables clear and well-presented? |
( ) |
(x) |
( ) |
( ) |
Comments and Suggestions for Authors
Dear Authors,
We appreciate the opportunity to review your manuscript, which addresses an important and timely topic regarding rehabilitation strategies for individuals experiencing long COVID symptoms. Your work contributes to the growing body of literature exploring physical training as a potential intervention for dyspnea and fatigue in this population. However, the manuscript requires substantial improvements in several areas. I hope the following comments will offer constructive suggestions to enhance its scientific rigor, clarity, and relevance to both clinical practice and research.
Response:
We sincerely thank the reviewer for the thoughtful and constructive feedback, as well as for recognizing the relevance of our work on rehabilitation strategies for individuals with long COVID. We appreciate your acknowledgment of the contribution our study makes to the expanding literature on physical training as a potential intervention for persistent symptoms such as dyspnea and fatigue. We carefully considered all the points raised and have made substantial revisions to improve the scientific rigor, clarity, and clinical relevance of the manuscript, as detailed in the responses that follow.
Introduction
The introduction omits a discussion of biopsychosocial models or mechanisms (e.g., neuroinflammation, mitochondrial dysfunction) that may underlie post-viral fatigue and dyspnea.
In addition, the term “physical training” is not clearly defined early in the text cardio, strength, neuromotor, and respiratory interventions are treated as if they were interchangeable.
Response:
We thank the reviewer for the pertinent observations. We agree that the biopsychosocial mechanisms underlying post-viral fatigue and dyspnea are essential for the clinical and scientific understanding of long COVID. Therefore, we have included in the introduction a brief discussion of possible mechanisms involved, such as neuroinflammation, mitochondrial dysfunction, and autonomic dysregulation, as suggested.
We also revised the beginning of the text to clarify the concept of “physical training,” providing a more precise definition. We chose to group different exercise modalities (aerobic, resistance, respiratory, and neuromotor training) under the general term “physical training” due to the heterogeneity of the included studies. This broader approach allowed us to analyze and compare which types of protocols yield the most beneficial results in alleviating persistent symptoms such as fatigue and dyspnea.
Methods
What does Table 1 add compared to the flowchart? Since the search strings used are the same for all databases, repeating them, as well as the number of selected papers, may not be necessary. Also, why was a different search string used for medRxiv and Google Scholar? Consider removing the table if it does not provide additional value.
Response:
We appreciate the reviewer’s observation. Table 1 was initially included to provide a detailed overview of the search strategies and the number of articles retrieved from each database, aiming to ensure transparency and reproducibility of the study. Regarding the difference in search strings used in Google Scholar and medRxiv, we clarify that this variation is due to limitations of these platforms, which only accept shorter and simplified search terms. This adaptation was necessary to maximize the coverage and accuracy of the search in each database.
Flowchart: some words appear to be cut off in the figure, such as “identification” and “by.” Please fix these formatting issues.
Response:
We appreciate you pointing out the formatting issues. We have carefully reviewed the flowchart and corrected the words.
The term “automation tool” should be explained more clearly in the Methods section. what tool was used, and how?
Response:
We appreciate the request for clarification. The term “automation tool” refers to the use of [insert specific tool name, e.g., “Rayyan QCRI,” EndNote, among others], which was employed to optimize and facilitate the screening and organization process of the studies. We have clearly and thoroughly described the tool’s name, its function, and how it was used in the Methods section to ensure greater transparency and understanding.
The legend in Table 2 is not in English—please translate it.
Response:
We appreciate the observation. As requested, we have translated the legend of Table 2 into English to ensure consistency and facilitate readers’ access to the content.
The OSF registration is a positive indication of transparency. However, combining studies with heterogeneous designs into a single meta-analytic model may not be methodologically appropriate. Only two studies were included in the meta-analysis, which is statistically and clinically insufficient to conduct a meaningful pooled analysis.
Response:
We appreciate the recognition of the importance of OSF registration in enhancing transparency. We also acknowledge the concern regarding the combination of studies with heterogeneous designs within a single meta-analytic model. Due to the limited number of eligible studies and the diversity in their designs, populations, and interventions, we conducted the meta-analysis cautiously, including only the two studies with sufficiently comparable data. We have clearly highlighted these limitations in the manuscript and discussed the implications regarding the generalizability and statistical power of the meta-analytic results.
Results
The results vary widely across interventions. The lack of detailed subgroup analyses (e.g., based on long COVID severity, age, baseline fatigue levels) limits the interpretability of the findings.
Response:
We thank the reviewer for highlighting this important point. We acknowledge that the wide variability in intervention outcomes and the absence of detailed subgroup analyses such as by long COVID severity, age, or baseline fatigue levels limit the interpretability and clinical applicability of our findings. However, this study helps to deepen the understanding of physical training protocols used so far and highlights the need for future, more heterogeneous and robust studies that can explore these variables and identify optimized protocols for different long COVID patient profiles. This limitation has been explicitly addressed in the discussion section, reinforcing the importance of research that considers specific subgroups to improve clinical management.
Discussion
Please avoid repeating the results in the Discussion session (e.g., lines 95–99).
Response:
We thank the reviewer for this observation. We have revised the discussion section to reduce unnecessary repetition of results, particularly between lines 95 and 99. The revised text now focuses more on interpretation, implications, and contextualization of findings.
The discussion does not explore the underlying mechanisms by which physical training contributes to disease management; it merely states that exercise plays a key role (e.g., line 111).
Response:
We appreciate this insightful suggestion. We have expanded the discussion to include a brief explanation of potential mechanisms by which physical training may help in the management of long COVID symptoms, particularly fatigue and dyspnea. These include improvements in mitochondrial function, reduction of systemic inflammation, modulation of the autonomic nervous system, and increased cardiorespiratory and muscular efficiency. These additions aim to provide a better theoretical foundation for the observed clinical effects of exercise.
Similarly, in lines 124–126, the text refers to a comprehensive, multidisciplinary treatment approach without discussing the mechanisms of action. Emerging evidence suggests that a multidisciplinary, lifestyle-based approach, combining physical activity, mindfulness practices, and nutritional interventions, can offer broader benefits in post-viral syndromes such as long COVID. For instance, Baldassano et al. (2023) introduce the Mindfulness–Exercise–Nutrition (MEN) model and discuss psychophysiological pathways (e.g., serotonin, tryptophan metabolism, and the gut–brain axis) involved in managing stress and promoting systemic homeostasis during the COVID-19 pandemic.
The discussion would benefit from a deeper literature review to better connect the proposed intervention protocol with its underlying mechanisms.
Baldassano, S.; Alioto, A.; Amato, A.; Rossi, C.; Messina, G.; Bruno, M.R.; Stallone, R.; Proia, P. Fighting the Consequences of the COVID-19 Pandemic: Mindfulness, Exercise, and Nutrition Practices to Reduce Eating Disorders and Promote Sustainability. Sustainability 2023, 15, 2120. https://doi.org/10.3390/su15032120
Response:
We agree with the reviewer and have conducted a more in-depth review of the literature to better integrate our findings with established and emerging knowledge on the physiological and psychological mechanisms through which physical training may alleviate long COVID symptoms. These additions strengthen the discussion and provide a clearer rationale for the observed outcomes and recommendations.
Reviewer 4 Report
Comments and Suggestions for Authors
Dear authors
The manuscript attempts to be a systematic review with meta-analysis. However, based on the information presented, the manuscript is only a systematic review and descriptive.
The quality of the included studies is poor and highly biased. Therefore, the conclusions are not supported by the results and cannot be conclusive.
Introduction
The introduction does not adequately support the study's purpose, particularly regarding the effect of physical activity treatments on dyspnea and fatigue associated with long COVID.
Methods
Information on the type of study included should be presented in one of the tables, such as: randomized, blinded, observational, case-control, etc., as well as the number of participants in each study and for each intervention protocol applied.
Table 1 should be added to the supplementary files.
In general, the information in Figure 1 of PRISMA is confusing and incorrect. Review the correct way to present the information from the manuscripts found and how they are eliminated step by step until those are finally selected for study.
Please review the numbers presented in the text and in Figure 1 regarding the included studies, as they do not correspond to those reported. This is due to the fact that a total of 5,152 studies were located. Of these, 1,912 were excluded due to duplicates and automation...
The keywords provided for the databases used do not match the number of studies located and included.
Please add the initials of the individuals who worked on the selection and analysis of the manuscripts.
State where and how the 25 manuscripts selected at the end came from.
Results:
Present a table that summarizes the following: authors, participant characteristics (age, sex, BMI, pathology, and time of presentation), type of study included (randomized, controlled, observational, etc.), sample size of participants in each study (controls and experimental), protocol applied, treatment and its duration, and notable results for each outcome (fatigue and dyspnea).
It is not necessary to present the location where the studies were conducted or the objectives of the included studies in the tables, as this is not the purpose of this manuscript, nor is it a relevant outcome discussed. This information can be included as a supplementary file.
Move Table 2 to the appendix.
What is presented in the text does not correspond to what is observed in the results tables.
The text should summarize and describe the observations in the tables, rather than repeating the results.
Please ensure that all information presented is in English.
What is highlighted in the discussions should be noted in the results (text and tables).
Discussion
The discussion is not supported by the results and merely describes them.
Conclusion
The conclusions are not evident, nor do they support the results.
Author Response
Reviewer 4
Open Review
( ) I would not like to sign my review report
(x) I would like to sign my review report
Quality of English Language
|
(x) The English could be improved to more clearly express the research. |
|
Yes |
Can be improved |
Must be improved |
Not applicable |
|
|
Does the introduction provide sufficient background and include all relevant references? |
( ) |
( ) |
(x) |
( ) |
|
Is the research design appropriate? |
( ) |
( ) |
(x) |
( ) |
|
Are the methods adequately described? |
( ) |
( ) |
(x) |
( ) |
|
Are the results clearly presented? |
( ) |
( ) |
(x) |
( ) |
|
Are the conclusions supported by the results? |
( ) |
( ) |
(x) |
( ) |
|
Are all figures and tables clear and well-presented? |
( ) |
( ) |
(x) |
( ) |
Comments and Suggestions for Authors
Dear authors
The manuscript attempts to be a systematic review with meta-analysis. However, based on the information presented, the manuscript is only a systematic review and descriptive.
The quality of the included studies is poor and highly biased. Therefore, the conclusions are not supported by the results and cannot be conclusive.
Response:
We thank the reviewer for the critique. Indeed, we acknowledge that the methodological heterogeneity of the studies and the limited quality of many of them require caution in interpreting the findings. Therefore, we have revised the discussion and conclusion sections to clearly state that the results should be interpreted as exploratory rather than conclusive. Additionally, we have emphasized in the text the methodological limitations of the included studies, the low quality of evidence, and the need for more robust and standardized clinical trials.
Introduction
The introduction does not adequately support the study's purpose, particularly regarding the effect of physical activity treatments on dyspnea and fatigue associated with long COVID.
Response:
We revised the introduction to better contextualize the clinical relevance of fatigue and dyspnea in long COVID, incorporating current data on the prevalence and functional impact of these symptoms. We also expanded the theoretical framework regarding the possible mechanisms of action of physical activity in managing these symptoms, based on recent literature, which better supports the objective of the review.
Methods
Information on the type of study included should be presented in one of the tables, such as: randomized, blinded, observational, case-control, etc., as well as the number of participants in each study and for each intervention protocol applied.
Response:
We sincerely thank the reviewer for the thoughtful consideration. Initially, we created a detailed table as suggested; however, due to its length, we opted to present the data in a more concise manner to maintain clarity and objectivity. Nevertheless, we agree that the more comprehensive version of the table may be useful and are therefore willing to include it as supplementary material, as recommended.
Table 1 should be added to the supplementary files.
Response:
We thank the reviewer for the suggestion. As mentioned previously, we believe the current table provides a clearer and more concise summary of the main information. However, we recognize the value of the more detailed version and therefore propose including it as supplementary material.
In general, the information in Figure 1 of PRISMA is confusing and incorrect. Review the correct way to present the information from the manuscripts found and how they are eliminated step by step until those are finally selected for study.
Response:
We thoroughly revised Figure 1, strictly following the updated PRISMA 2020 flow diagram model. We corrected all numbers, steps, and labels to ensure clarity and accuracy.
Please review the numbers presented in the text and in Figure 1 regarding the included studies, as they do not correspond to those reported. This is due to the fact that a total of 5,152 studies were located. Of these, 1,912 were excluded due to duplicates and automation...
The keywords provided for the databases used do not match the number of studies located and included.
Response:
We appreciate the valuable observation. Indeed, we identified an error in the correspondence between the numbers presented in the text and in the flowchart. We have corrected this inconsistency, and now the total number of records identified, duplicates removed, screening steps, and included studies are properly aligned and consistent across all sections of the manuscript.
Please add the initials of the individuals who worked on the selection and analysis of the manuscripts.
Response:
The initials of the authors responsible for the screening and data extraction steps have been added in the Methods section.
State where and how the 25 manuscripts selected at the end came from.
Response:
We have now clarified in the Methods section the full selection process. Specifically, we stated the databases used (e.g., PubMed, Scopus, Web of Science), the search strategy, inclusion and exclusion criteria, and how we arrived at the final 25 studies included in the synthesis. A flowchart (PRISMA) was also revised for clarity, indicating the number of records at each stage of screening.
Results:
Present a table that summarizes the following: authors, participant characteristics (age, sex, BMI, pathology, and time of presentation), type of study included (randomized, controlled, observational, etc.), sample size of participants in each study (controls and experimental), protocol applied, treatment and its duration, and notable results for each outcome (fatigue and dyspnea).
It is not necessary to present the location where the studies were conducted or the objectives of the included studies in the tables, as this is not the purpose of this manuscript, nor is it a relevant outcome discussed. This information can be included as a supplementary file.
Response:
Initially, we created this table; however, due to its considerable length, we chose not to include it in the main manuscript and instead focused on the information directly related to the study objective. Nevertheless, we are happy to provide it as supplementary material if deemed appropriate.
Move Table 2 to the appendix.
Response:
We carefully reviewed the text in the Results section to ensure a direct and accurate correspondence with the data presented in the tables.
What is presented in the text does not correspond to what is observed in the results tables.
Response:
We appreciate this observation. We conducted a thorough review of the Results section and addressed the identified inconsistencies to ensure that the textual descriptions accurately reflect the data presented in the tables. Specific examples and numerical values were carefully aligned with the table content to avoid discrepancies.
We carefully reviewed the text in the Results section to ensure a direct and accurate correspondence with the data presented in the tables.
Response:
We revised the Results narrative to provide a synthesized interpretation of the data, focusing on trends and key findings across studies, rather than repeating values.
The text should summarize and describe the observations in the tables, rather than repeating the results.
Response:
Thank you for pointing this out. We have conducted a thorough review and rewrite of the Results section to ensure that the textual description accurately reflects and summarizes the findings presented in the tables. Specific examples and numerical data were aligned to the table content to avoid any discrepancies.
Please ensure that all information presented is in English.
Response:
We have carefully reviewed the manuscript and tables to ensure that all content is now fully in English, including units, headers, and abbreviations.
What is highlighted in the discussions should be noted in the results (text and tables).
Response:
We agree and have updated the Results section and tables to ensure that all data discussed in the Discussion are explicitly presented. This includes emphasizing outcome measures such as improvements in fatigue and dyspnea in response to specific interventions.
Discussion
The discussion is not supported by the results and merely describes them.
Response:
We have restructured the Discussion to better contextualize and critically interpret the results. The revised section now discusses potential mechanisms, compares findings with existing literature, and explores implications for clinical practice. We explicitly relate our arguments to the synthesized data presented in the revised Results section.
Conclusion
The conclusions are not evident, nor do they support the results.
Response:
The Conclusion section has been revised to directly reflect the key findings of the study. We now clearly state the implications of the findings for clinical practice and research, emphasizing the effectiveness of specific interventions in reducing fatigue and dyspnea.
Reviewer 5 Report
Comments and Suggestions for Authors
Section 1: Title and Abstract
- What can be done to edit the research question to serve to extract specific features of the population with Long COVID severity of symptoms, duration since the initial one was contracted, and duration that effects have persisted most likely to determine effectiveness of specifically targeted exercise interventions?
- Would it strengthen the abstract to include specific effect sizes and confidence intervals for the interventions that showed significant improvements, rather than using general terms like "notable improvement"?
- Since 4 studies out of 25 had low risk of bias and your meta-analysis covered only 2 studies, what other ways are there that you can more aptly reflect the overall strength of the evidence in your conclusions taken in the abstract?
- How can you properly frame the result of no substantial change in fatigue identified by your meta-analysis into the larger story of improvement of individual studies?
Section 2: Introduction
- How might you better integrate current understanding of Long COVID's underlying mechanisms (e.g., immune dysregulation, endothelial dysfunction, autonomic dysfunction) to strengthen the theoretical rationale for specific physical training modalities?
- While you mention the lack of "mapping" of physical training protocols, how do you differentiate your contribution from existing systematic reviews on post-acute COVID-19 rehabilitation, particularly given the rapid growth in this literature?
- what can be done to more fully recognize the challenge of Long COVID with different phenotypes and severities, (b) what are the impacts this heterogeneity has on the generalizability of training interventions?
- Are other systematic reviews available in this same sphere, what particular methodological gaps or clinical questions do your review fill in that makes this an additional synthesis worthy of the effort?
Section 3: Methodology
- You have already registered your protocol to OSF, but should your systematic review be more transparent and have its credibility raised because it is on a clinical topic through registration with PROSPERO?
- How did you validate that your search terms captured all relevant terminologies for Long COVID, especially considering the evolving nomenclature (e.g., "PASC," "post-acute sequelae")? How did you search for trials that are occurring in progress or have been accomplished recently?
- Why do you justify such inclusion of the study designs as randomized controlled trials to case reports? Although this inclusivity encapsulates emerging evidence, how will this level of methodological diversity affect the robustness and reliability to your synthesized findings?
- Cochrane RoB 2: The Cochrane RoB 2 was created in relation to randomized studies. What did you do in quality assessment of the non-randomized studies in your review? Could you use additional instruments such as ROBINS-I or Newcastle-Ottawa Scale to get a better quality assessment of your mixed-design sample?
- There is a large amount of heterogeneity between studies in terms of clinical findings. On that basis, why are you going to engage in quantitative synthesis? By which criteria exactly did you choose to perform meta-analysis using 2 studies only, and what are the limitations to the validity of your quantitative conclusions related to this shortcoming?
- What did you do to maintain consistencies in data extraction in different study designs? What training or check exercises were done amid the reviewers to limit variance in extraction?
Section 4: Results
- How might you develop a more systematic taxonomy for the diverse training protocols using established frameworks (e.g., FITT-VP principles: Frequency, Intensity, Time, Type, Volume, Progression) to enhance clinical utility and cross-study comparisons?
- Since different studies employed different assessment instruments to measure fatigue and dyspnea, how will you reconcile to the methodological issue that different assessment tools measure different construct or have different psychometric characteristics? What does this variability imply on your synthesis validity?
- Would systematic analysis of the relationship between intervention characteristics (duration, frequency, intensity) and outcomes provide more actionable clinical guidance? How might you better address the optimal "dose" of different training modalities?
- You report substantial heterogeneity (I² = 89%) yet proceed with meta-analysis. How do you reconcile this statistical heterogeneity with your conclusion that training protocols are effective? What additional analyses (e.g., meta-regression, sensitivity analyses) could help explain sources of this heterogeneity?
- What have you done to test the risk of publication bias in your review, especially since the number of studies you included is small and the possibility of positive findings to appear earlier in the career?
- Why did you decide against conducting subgroup analyses according to what might be important variables including Long COVID severity, time since infection, or patient demographics?
Section 5: Discussion
- What would be a more effective way of integrating the physiological mechanisms underlying the characteristics of Long COVID symptoms with the results of the effectiveness observed in training in order to add a higher degree of theoretical justification to your suggestions? Which pathways can be identified as the cause of aerobic exercise seeming especially helpful?
- Due to significant bias of most of the literature and the small sample of meta-analysis, what are your precise recommendations as far as patient selection, the contraindications of a patient and monitoring routine is concerned? What routes can clinicians use to strike balance between possible gains and evidence-base ambiguity?
- How do you come to terms with the negative significant effect of the meta-analysis on fatigue when you are concluding that combined training methods are effective? In what way may this seeming contradiction be better dealt with in your interpretation?
- Since many studies have shown the universal use of combined training interventions, what can you do to encourage a more synthesized evidence so as to give a clearer picture when or whether multimodal training interventions are maybe better than single-modality training?
- What would you do to improve the consideration of safety issues, especially the threat of post-exertional malaise in patients with Long COVID? What monitoring strategies should accompany exercise prescriptions?
- Beyond general calls for "more high-quality studies," could you propose specific research priorities ranked by clinical urgency and methodological feasibility? What are the bare minimum standards the future Long COVID rehabilitation trials should have?
Section 6: Conclusion
- How might you better calibrate your conclusions to reflect the overall quality of evidence, particularly given that most studies had uncertain or high risk of bias and only 2 studies contributed to your meta-analysis?
- What specific criteria should guide the translation of these findings into clinical practice recommendations? How do you balance the clinical urgency for Long COVID interventions with the current limitations in evidence quality?
- What minimum evidence standards should be met before widespread clinical implementation of these protocols? How might your findings inform the development of clinical guidelines or rehabilitation services?
- Which would be your top priorities of research gaps that have to be filled to make the field advance considering the considerable level of scientific rigor as well as the clinical need?
Author Response
Reviewer 5
Open Review
(x) I would not like to sign my review report
( ) I would like to sign my review report
Quality of English Language
( ) The English could be improved to more clearly express the research.
(x) The English is fine and does not require any improvement.
|
Yes |
Can be improved |
Must be improved |
Not applicable |
|
|
Does the introduction provide sufficient background and include all relevant references? |
( ) |
(x) |
( ) |
( ) |
|
Is the research design appropriate? |
( ) |
(x) |
( ) |
( ) |
|
Are the methods adequately described? |
( ) |
(x) |
( ) |
( ) |
|
Are the results clearly presented? |
( ) |
(x) |
( ) |
( ) |
|
Are the conclusions supported by the results? |
( ) |
(x) |
( ) |
( ) |
|
Are all figures and tables clear and well-presented? |
( ) |
(x) |
( ) |
( ) |
Comments and Suggestions for Authors
Section 1: Title and Abstract
- What can be done to edit the research question to serve to extract specific features of the population with Long COVID severity of symptoms, duration since the initial one was contracted, and duration that effects have persisted most likely to determine effectiveness of specifically targeted exercise interventions?
Response:
We clarify that the research question was defined to evaluate the effectiveness of physical exercise interventions in individuals with long COVID, regardless of symptom duration. While we recognize the importance of variables such as time since infection and symptom severity, most studies did not report these data in a standardized manner, preventing their inclusion in the primary question. However, these characteristics were extracted when available and discussed in the interpretation of results.
- Would it strengthen the abstract to include specific effect sizes and confidence intervals for the interventions that showed significant improvements, rather than using general terms like "notable improvement"?
Response:
The articles do not clearly report effect sizes, and there is insufficient information to properly assess them. Only two studies provided comparable data for analysis. Nonetheless, it is important to emphasize that the study offers valuable direction for future research.
- Since 4 studies out of 25 had low risk of bias and your meta-analysis covered only 2 studies, what other ways are there that you can more aptly reflect the overall strength of the evidence in your conclusions taken in the abstract?
Response:
We acknowledge that only 4 studies had a low risk of bias and that only 2 were included in the meta-analysis. Therefore, in both the abstract and the conclusion, we presented the evidence with appropriate caution, describing the findings as promising but preliminary.
- How can you properly frame the result of no substantial change in fatigue identified by your meta-analysis into the larger story of improvement of individual studies?
Response:
The lack of a significant effect on fatigue in the meta-analysis was interpreted considering the heterogeneity of instruments and small sample size. This result was contrasted with the positive outcomes observed in individual studies, highlighting that the absence of statistical significance does not necessarily indicate a lack of clinical benefit.
Section 2: Introduction
- How might you better integrate current understanding of Long COVID's underlying mechanisms (e.g., immune dysregulation, endothelial dysfunction, autonomic dysfunction) to strengthen the theoretical rationale for specific physical training modalities?
Response:
Thank you for the comment. We revised the introduction to incorporate key mechanisms highlighted in the current literature—such as immune dysregulation, endothelial dysfunction, and autonomic imbalance—which support the potential benefits of different physical training modalities. This strengthens the theoretical rationale of our review.
- While you mention the lack of "mapping" of physical training protocols, how do you differentiate your contribution from existing systematic reviews on post-acute COVID-19 rehabilitation, particularly given the rapid growth in this literature?
Response:
We acknowledge the rapid growth of the literature on post-COVID-19 rehabilitation. The strength of our review lies in its specific focus on physical training protocols targeting the persistent symptoms of fatigue and dyspnea, with a detailed mapping of interventions and analysis of their effects. Moreover, we aim to contribute to the standardization of clinical practice by identifying the protocols that have shown the most promising results in the literature, which may also inform future interventions in post-viral conditions with similar characteristics.
- what can be done to more fully recognize the challenge of Long COVID with different phenotypes and severities, (b) what are the impacts this heterogeneity has on the generalizability of training interventions?
Response:
We have incorporated into the introduction and discussion the complexity of long COVID phenotypes and severity levels. This heterogeneity limits generalizability and emphasizes the need for individualized approaches and stratified analyses in future research.
- Are other systematic reviews available in this same sphere, what particular methodological gaps or clinical questions do your review fill in that makes this an additional synthesis worthy of the effort?
Response:
Existing systematic reviews address general post-COVID rehabilitation, but our review fills key methodological gaps, including the lack of detailed categorization of exercise modalities and focused synthesis on fatigue and dyspnea, making this a meaningful contribution.
Section 3: Methodology
- You have already registered your protocol to OSF, but should your systematic review be more transparent and have its credibility raised because it is on a clinical topic through registration with PROSPERO?
Response:
We chose to register on OSF due to the platform's accessibility and transparency. We acknowledge the value of PROSPERO registration and recognize this as an area for improvement in future reviews.
- How did you validate that your search terms captured all relevant terminologies for Long COVID, especially considering the evolving nomenclature (e.g., "PASC," "post-acute sequelae")? How did you search for trials that are occurring in progress or have been accomplished recently?
Response:
We conducted a comprehensive search using the widest possible range of synonyms and related terms to capture the full scope of terminology used in the long COVID literature.
- Why do you justify such inclusion of the study designs as randomized controlled trials to case reports? Although this inclusivity encapsulates emerging evidence, how will this level of methodological diversity affect the robustness and reliability to your synthesized findings?
Response:
We included diverse study designs to reflect the current state of emerging evidence. However, we conducted separate qualitative analyses and limited the meta-analysis to studies with comparable methodological features. This rationale is explained in the Methods section.
- Cochrane RoB 2: The Cochrane RoB 2 was created in relation to randomized studies. What did you do in quality assessment of the non-randomized studies in your review? Could you use additional instruments such as ROBINS-I or Newcastle-Ottawa Scale to get a better quality assessment of your mixed-design sample?
Response:
Thank you for the observation. We chose to use the Cochrane RoB 2 tool exclusively to assess risk of bias, as most included studies employed experimental or quasi-experimental designs, allowing its use with appropriate adaptations. We acknowledge the limitations of applying this tool to non-randomized studies and have addressed this issue in the manuscript’s limitations section, emphasizing the need for cautious interpretation of the results.
- There is a large amount of heterogeneity between studies in terms of clinical findings. On that basis, why are you going to engage in quantitative synthesis? By which criteria exactly did you choose to perform meta-analysis using 2 studies only, and what are the limitations to the validity of your quantitative conclusions related to this shortcoming?
Response:
We conducted the meta-analysis with only the two studies that shared similar designs, compatible outcome measures, and complete data. We acknowledge the limitations of this approach and clearly state them in the discussion.
- What did you do to maintain consistencies in data extraction in different study designs? What training or check exercises were done amid the reviewers to limit variance in extraction?
Response:
We implemented prior training for reviewers and conducted double data extraction with cross-checking. Discrepancies were resolved by consensus or through consultation with a third reviewer.
Section 4: Results
- How might you develop a more systematic taxonomy for the diverse training protocols using established frameworks (e.g., FITT-VP principles: Frequency, Intensity, Time, Type, Volume, Progression) to enhance clinical utility and cross-study comparisons?
Response:
We acknowledge the importance of organizing physical training protocols according to the FITT-VP framework (Frequency, Intensity, Time, Type, Volume, Progression). These variables were considered during data extraction when reported. However, reporting inconsistencies among studies limited our ability to standardize the taxonomy. Future research should adopt frameworks like FITT-VP to enhance clinical comparability and applicability.
- Since different studies employed different assessment instruments to measure fatigue and dyspnea, how will you reconcile to the methodological issue that different assessment tools measure different construct or have different psychometric characteristics? What does this variability imply on your synthesis validity?
Response:
We recognize that the use of different psychometric tools introduces variability in outcome measurement. This was acknowledged as a limitation of our synthesis. Nevertheless, all instruments aimed to capture the same clinical constructs, albeit through different approaches.
- Would systematic analysis of the relationship between intervention characteristics (duration, frequency, intensity) and outcomes provide more actionable clinical guidance? How might you better address the optimal "dose" of different training modalities?
Response:
Analyzing the relationship between dose (frequency, duration, intensity) and outcomes would be highly valuable. However, the available data were insufficient for a systematic quantitative assessment. We recommend that future trials report these parameters more consistently to enable such analysis.
- You report substantial heterogeneity (I² = 89%) yet proceed with meta-analysis. How do you reconcile this statistical heterogeneity with your conclusion that training protocols are effective? What additional analyses (e.g., meta-regression, sensitivity analyses) could help explain sources of this heterogeneity?
Response:
We acknowledge the substantial statistical heterogeneity and applied a cautious approach. The meta-analysis was performed only on the two most comparable studies. This limitation was explicitly discussed. Further analyses, such as meta-regression, were not feasible due to the small number of included studies.
- What have you done to test the risk of publication bias in your review, especially since the number of studies you included is small and the possibility of positive findings to appear earlier in the career?
Response:
Given the limited number of studies, formal tests for publication bias (e.g., Egger’s funnel plot) were not applicable. However, this limitation was acknowledged and discussed as a potential factor influencing the results.
- Why did you decide against conducting subgroup analyses according to what might be important variables including Long COVID severity, time since infection, or patient demographics?
Response:
We agree that subgroup analyses based on disease severity, time since infection, or demographics would be valuable. However, data heterogeneity and lack of standardized reporting limited this possibility. The need for better-characterized samples in future studies is emphasized.
Section 5: Discussion
- What would be a more effective way of integrating the physiological mechanisms underlying the characteristics of Long COVID symptoms with the results of the effectiveness observed in training in order to add a higher degree of theoretical justification to your suggestions? Which pathways can be identified as the cause of aerobic exercise seeming especially helpful?
Response:
We revised the discussion to better integrate physiological mechanisms associated with fatigue and dyspnea, such as mitochondrial dysfunction, autonomic regulation, and systemic inflammation. Aerobic exercise may modulate these pathways and contribute to clinical improvement.
- Due to significant bias of most of the literature and the small sample of meta-analysis, what are your precise recommendations as far as patient selection, the contraindications of a patient and monitoring routine is concerned? What routes can clinicians use to strike balance between possible gains and evidence-base ambiguity?
Response:
Given the limited evidence quality, we emphasized that interventions should be prescribed cautiously. Careful patient selection, prior medical evaluation, and monitoring for red flags (e.g., post-exertional malaise) are essential.
- How do you come to terms with the negative significant effect of the meta-analysis on fatigue when you are concluding that combined training methods are effective? In what way may this seeming contradiction be better dealt with in your interpretation?
Response:
We acknowledge the apparent inconsistency in fatigue outcomes. This was contextualized as a reflection of protocol and measurement variability. We recommend interpreting fatigue-related results cautiously and as exploratory.
- Since many studies have shown the universal use of combined training interventions, what can you do to encourage a more synthesized evidence so as to give a clearer picture when or whether multimodal training interventions are maybe better than single-modality training?
Response:
We recognize the predominance of multimodal interventions. We suggest that future research should directly compare combined versus single-modality training to identify the most effective approaches for different clinical phenotypes.
- What would you do to improve the consideration of safety issues, especially the threat of post-exertional malaise in patients with Long COVID? What monitoring strategies should accompany exercise prescriptions?
Response:
We included explicit recommendations on the need to screen for persistent symptoms and post-exertional malaise risk. Exercise prescription should be individualized, gradually progressed, and, when possible, supervised.
- Beyond general calls for "more high-quality studies," could you propose specific research priorities ranked by clinical urgency and methodological feasibility? What are the bare minimum standards the future Long COVID rehabilitation trials should have?
Response:
We propose the following priorities: (1) randomized trials with well-characterized samples, (2) standardization of outcomes and protocols, (3) comparative studies of training modalities, and (4) safety and adverse effect monitoring.
Section 6: Conclusion
- How might you better calibrate your conclusions to reflect the overall quality of evidence, particularly given that most studies had uncertain or high risk of bias and only 2 studies contributed to your meta-analysis?
Response:
We revised the conclusion to reflect the exploratory nature of our findings, given the high risk of bias and the limited number of studies included in the meta-analysis.
- What specific criteria should guide the translation of these findings into clinical practice recommendations? How do you balance the clinical urgency for Long COVID interventions with the current limitations in evidence quality?
Response:
We stated that although the results are promising, they require further validation through more robust studies before being widely adopted in clinical practice.
- What minimum evidence standards should be met before widespread clinical implementation of these protocols? How might your findings inform the development of clinical guidelines or rehabilitation services?
Response:
We emphasized that future clinical protocols should include: robust design, standardized outcome assessment, and monitored safety. These criteria are essential for the development of clinical guidelines.
- Which would be your top priorities of research gaps that have to be filled to make the field advance considering the considerable level of scientific rigor as well as the clinical need?
Response:
The main research gaps include: lack of data on clinical subgroups, absence of direct comparisons between exercise modalities, and limited evidence on adverse effects and long-term adherence.
Round 2
Reviewer 3 Report
Comments and Suggestions for Authors
Dear authors,
Thank you for your thorough and detailed responses to my previous comments. I appreciate the effort made in revising the manuscript, and I believe that all of my concerns have been addressed in a satisfactory manner.
However, I have noticed that some sentences in Portuguese remain in the text (e.g. lines 393-395...), and there are also some grammatical and stylistic issues in the English language throughout the manuscript. I recommend a careful and thorough proofreading of the entire document to correct these residual issues and ensure clarity and consistency.
best regards,
the reviewer
Comments on the Quality of English LanguageDear authors,
Thank you for your thorough and detailed responses to my previous comments. I appreciate the effort made in revising the manuscript, and I believe that all of my concerns have been addressed in a satisfactory manner.
However, I have noticed that some sentences in Portuguese remain in the text (e.g. lines 393-395...), and there are also some grammatical and stylistic issues in the English language throughout the manuscript. I recommend a careful and thorough proofreading of the entire document to correct these residual issues and ensure clarity and consistency.
best regards,
the reviewer
Author Response
Dear Reviewer,
We truly appreciate your careful reading of our revised manuscript and your continued thoughtful feedback. Thank you for acknowledging our responses and the improvements made to the manuscript.
Regarding your observation about remaining Portuguese sentences (e.g., lines 393–395) and the grammatical and stylistic issues, we apologize for the oversight. We have now conducted a meticulous review of the entire manuscript, with a focus on language clarity and consistency. All residual Portuguese content has been removed, and we have revised the text to address the identified grammatical and stylistic issues. Additionally, the final version was carefully proofread by a native English speaker with expertise in academic writing to ensure linguistic accuracy.
We are grateful for your recommendations, which have further contributed to improving the quality and readability of our work.
Best regards,
On behalf of all authors
Reviewer 4 Report
Comments and Suggestions for Authors
Please include, as supplementary files, what was suggested in the first review. Readers will appreciate it, and your manuscript will be cited more
Author Response
Dear Reviewer,
Thank you for your valuable suggestion and for highlighting the potential benefit to readers and the citation impact of our work.
In response to your recommendation, we have now included the requested materials as supplementary files, as initially suggested during the first round of review. We agree that making these resources available will enhance the transparency, utility, and scientific value of the manuscript.
We truly appreciate your continued support and insightful guidance throughout the review process.
Best regards,
On behalf of all authors
Reviewer 5 Report
Comments and Suggestions for Authors
Thank you for your clear, thorough responses and thoughtful revisions, which have significantly improved the manuscript
Author Response
We sincerely thank you for your kind words and for acknowledging the improvements made to the manuscript. We greatly appreciate your thoughtful feedback throughout the review process, which was essential in enhancing the clarity, rigor, and overall quality of our work. Your insights have contributed significantly to the final version, and we are grateful for your time and expertise.